# Function-Valued Causal Influence in Nonlinear Time Series

Valentina V. Kuskova [1]   Dmitry Zaytsev [1]   Michael Coppedge [2]

## Abstract

Causal discovery in time series is increasingly performed using nonlinear machine-learning models, yet the resulting causal relationships are almost always summarized by scalar edge scores. We argue that this practice obscures the true object learned by nonlinear autoregressive models: a state-dependent function whose effect varies across regimes, magnitudes, and contexts. We formalize function-valued causal influence for additive, contribution-decomposable architectures and show that scalar causal scores constitute a severe information bottleneck, conflating between-state variation with within-state residual noise. Using Neural Additive Vector Autoregression as a representative architecture, we introduce a practical framework based on Individual Conditional Expectation for estimating causal response functions directly from trained models. Through controlled synthetic experiments, we demonstrate that edges with indistinguishable scalar scores can exhibit qualitatively different functional behaviors, including monotonic, thresholded, saturating, and sign-changing effects. An applied case study on democratic development further shows that function-valued analysis reveals regime-specific and asymmetric causal structure systematically missed by score-centric approaches.

## 1. Introduction

Nonlinear causal discovery in time series has advanced rapidly in recent years, driven by flexible machine-learning models that learn rich, state-dependent representations of temporal dependence (Chen et al., 2025). Neural autoregressive and additive architectures extend classical vector autoregression (Geiger et al., 2015) by allowing causal re-

[1]Lucy Family Institute for Data & Society, University of Notre Dame, Notre Dame, Indiana, USA. [2]Department of Political Science, University of Notre Dame, Notre Dame, Indiana, USA. Correspondence to: Valentina V. Kuskova <vkuskova@nd.edu>.

*Proceedings of the 43rd International Conference on Machine Learning*, Seoul, South Korea. PMLR 306, 2026. Copyright 2026 by the author(s).

lationships to vary across states of the system (Bussmann et al., 2021), enabling the representation of threshold effects, saturation, asymmetry, and other nonlinear dynamics that are inaccessible to linear methods (Assaad et al., 2022). As a result, modern causal time-series models are capable of learning expressive, structured representations of directed temporal influence (Liao et al., 2025).

Despite these advances, the outputs of nonlinear causal discovery methods are almost universally reported in a highly compressed form (Montagna et al., 2023). Causal influence is typically summarized using scalar edge scores that quantify the strength or importance of directed relationships between variables (Tramontano et al., 2024). These scores are convenient for ranking edges, thresholding graphs, and benchmarking performance (Job et al., 2025), and they mirror the role played by coefficients in linear vector autoregression (Ahn, 1997). However, this practice introduces a fundamental mismatch between the representations learned by nonlinear causal models and the summaries used to evaluate and interpret them.

In nonlinear systems, causal influence cannot, in general, be represented by a single magnitude (Wan et al., 2025). Nonlinear autoregressive models do not learn fixed effects or constant weights; instead, they learn state-dependent functions that map past system configurations to future outcomes (Esarey & DeMeritt, 2014). Each directed relationship in such a model therefore corresponds not to a scalar quantity, but to a function-valued causal influence whose effect can vary across regimes, magnitudes, and contexts of the system (Troster & Wied, 2021). Reducing these functions to scalar summaries collapses essential structure and can obscure threshold behavior, saturation, asymmetry, and sign changes that are intrinsic to nonlinear dynamics (Wan et al., 2025).

This representational collapse has important consequences for the adequacy of model outputs as summaries of learned causal structure, even when predictive accuracy is high. Scalar summaries average over regimes and interactions, masking where and how causal influence operates (Seitzer et al., 2021). Two causal relationships with similar scalar scores may correspond to qualitatively different mechanisms, while strong but localized effects may appear indistinguishable from weak but globally active ones (Rottman & Hastie, 2014). As a result, scalar causal scores provide, at best, a partial and potentially misleading picture of causal

structure in nonlinear time series.

To provide an example, consider two directed relationships in a democracy panel. Suppose a nonlinear model learns that civil liberties influence electoral quality, but only after a moderate institutional baseline is crossed, with no effect below that threshold. A second relationship captures the influence of judicial constraints on electoral quality through a monotonically linear effect of similar overall magnitude. Both edges receive comparable scalar causal scores of, e.g., approximately 0.15. Yet their policy implications are entirely different: the first implies a threshold below which interventions are unlikely to improve electoral outcomes; the second implies a uniform marginal return across all levels of judicial strength. Scalar summaries cannot communicate this distinction. Function-valued causal influence, as we formalize and estimate in this paper, recovers exactly this structure, and we demonstrate empirically in Section 5.2 that this scenario arises in real democratic panel data.

The consequences of this mismatch are particularly salient in theory-driven domains, such as the social sciences, which serve as a stringent stress test for causal representations (Sobel, 2000). In these settings, causal priorities must be interpretable and defensible (Hofman et al., 2021) in terms of mechanisms and regimes, not merely detected or ranked. Models that cannot explain how causality operates (e.g., when effects activate, saturate, or reverse), are rarely adopted in substantive research or policy analysis, regardless of predictive performance (de Slegte et al., 2025; Gerring, 2010). Consequently, many nonlinear causal models remain underutilized (De Bruijn et al., 2022) in precisely the domains where their expressive power would be most valuable.

In this paper, we address this gap by reframing causal influence in nonlinear time-series models as a function-valued object rather than a scalar score. We formalize function-valued causal influence for a broad class of additive, contribution-decomposable autoregressive models, independent of a specific architecture, and show that commonly used scalar causal scores correspond to low-dimensional projections of these functions, constituting an information bottleneck. We then introduce a practical, intervention-style framework based on Individual Conditional Expectation (ICE) (Goldstein et al., 2015) for estimating causal response functions directly from trained models. Through controlled synthetic systems, we demonstrate that qualitatively distinct causal mechanisms can collapse to similar scalar summaries. We further show, using a panel dataset on democratic development, that function-valued analysis reveals regime-dependent causal behavior that is systematically invisible to scalar scores.

## 2. Background and Setup

### 2.1. Multivariate Time Series and Autoregressive Prediction

We consider a multivariate time series $\{X_t\}_{t=1}^T$, where each observation $x \in \mathbb{R}^N$ is a vector of $N$ variables measured at time $t$. Let $K \in \mathbb{N}$ denote a fixed lag window. Autoregressive models predict the current state $X_t$ using its recent history

$$X_{t-K:t-1} = \{X_{t-K}, ..., X_{t-1}\}$$

thereby capturing temporal dependence through lagged inputs. An autoregressive predictor for the $j$-th variable takes the general form

$$\hat{X}_t^j = f_j(X_{t-K:t-1}),$$

where $f_j(\cdot)$ is an arbitrary, potentially nonlinear function. Throughout this paper, we focus on models trained to minimize a predictive loss (e.g., squared error). Under this setting, causal interpretation follows the standard Granger-style notion of predictive influence (Zhou et al., 2024; Zhang et al., 2020) in which causality is defined operationally via improvements in prediction.

### 2.2. Directed Predictive Influence and Causal Edges

In this framework, we define a directed edge $i \rightarrow j$ to mean that the lagged history of variable i contributes to predicting variable j. Formally, an edge $i \rightarrow j$ exists if the prediction function $f_j(\cdot)$ depends on $X_{t-K:t-1}^{(i)}$ in a way that improves predictive accuracy. This notion aligns with Granger causality (Zhou et al., 2024; Zhang et al., 2020): a variable is said to have a causal influence if its past helps predict another variable's future beyond what is achievable using other variables alone.

Importantly, this definition does not imply structural or interventional causality in the sense of causal graphs or counterfactual frameworks. Rather, it captures directed predictive influence in time series, which is the operational notion used by most nonlinear Granger-causal discovery methods.

### 2.3. Additive and Contribution-Decomposable Autoregressive Models

Our analysis focuses on a broad class of autoregressive models that are additive over variables (Li & Genton, 2009), allowing predictions to be decomposed into per-variable contributions. Specifically, for each target variable $j$, the predictor takes the form

$$\hat{X}_t^{(j)} = \beta_j + \sum_{i=1}^N f_{ij}\left(X_{t-K:t-1}^{(i)}\right)$$

where each $f_j(\cdot)$ models the influence of the lagged history of variable ion target $j$. This additive structure ensures that the contribution of each source variable can be isolated and analyzed separately.

Such models include linear vector autoregressive models (VAR) as a special case, where each $f_{ij}$ is linear, as well as a variety of nonlinear extensions (Kolesár & Plagborg-Møller, 2025) in which $f_{ij}$ is parameterized by flexible function approximators (e.g., neural networks or spline-based models). The key property for our purposes is contribution decomposability (Mediano et al., 2022): for any time $t$, the predicted value $\hat{X}_t^{(j)}$ can be written as a sum of identifiable source-specific terms.

Neural Additive Vector Autoregression (NAVAR) (Bussmann et al., 2021) is one representative instantiation of this model class, using neural networks to parameterize the functions $f_{ij}$ while enforcing additivity. We use NAVAR in our experiments due to its flexibility and interpretability, but our analysis applies to any autoregressive model that admits a similar additive decomposition. Implementation details specific to NAVAR are provided in the Appendix A.

## 2.4. Scope and Interpretation

Throughout the paper, we interpret causal influence in the predictive, Granger-style sense described above (Zhang et al., 2020; Zhou et al., 2024). Our goal is not to identify structural causal effects or to make counterfactual claims about interventions in the underlying data-generating process. Instead, we analyze what nonlinear autoregressive models learn about directed predictive relationships and how those relationships are represented and summarized. In particular, we study the representational consequences of summarizing the learned functions $f_{ij}(\cdot)$ using scalar edge scores, and we ask whether such summaries faithfully capture the information encoded by nonlinear, state-dependent causal influences.

## 3. Function-Valued Causal Influence

Nonlinear additive autoregressive models learn functions, not scalar effects (Li et al., 2024). However, in practice their outputs are almost always summarized using scalar edge scores (Rogovchenko et al., 2024). In this section, we formalize the appropriate causal object learned by such models - function-valued causal influence - and show why scalar summaries constitute a severe representational bottleneck.

**Definition**. Consider an additive autoregressive model of the form introduced in Section 2, where $f_{ij}(\cdot)$ denotes the contribution of the lagged history of variable $i$ to predicting target variable $j$. For any time $t$, define the instantaneous contribution of variable $i$ to variable $j$ as

$$\text{contrib}_{ij,t} \;\equiv\; f_{ij}\!\left(X_{t-K:t-1}^{(i)}\right).$$

so that the prediction is decomposed additively across sources.

**Function-valued influence (marginal form)**. The simplest function-valued causal influence object for edge i→j is the conditional expectation of this contribution given the most recent lag of the source variable:

$$g_{ij}(x) \;=\; \mathbb{E}\!\left[\text{contrib}_{ij,t} \;\middle|\; X_{t-1}^{(i)} = x\right].$$

This function characterizes how the expected contribution of source $i$ to target $j$ varies as a function of the state $x$ of the source variable. Importantly, $g_{ij}(\cdot)$ is generally nonlinear and state-dependent, even when averaged over all other variables and lags.

**Conditional and regime-dependent extensions.** Richer versions of function-valued influence can be defined by conditioning on additional variables. For example, conditioning on the target's recent history yields:

$$g_{ij}(x, y) \;=\; \mathbb{E}\!\left[\text{contrib}_{ij,t} \;\middle|\; X_{t-1}^{(i)} = x, \; X_{t-1}^{(j)} = y\right].$$

More generally, one may condition on a regime variable $r_t$, such as a summary index or a discretized state of the system:

$$g_{ij}(x \mid r) \;=\; \mathbb{E}\!\left[\text{contrib}_{ij,t} \;\middle|\; X_{t-1}^{(i)} = x, \; r_t = r\right].$$

These conditional definitions allow causal influence to vary across regimes, capturing asymmetries, thresholds, and state-specific activation patterns that are common in nonlinear systems.

**Relationship to scalar causal scores.** Most existing methods reduce the edge $i \to j$ to a scalar causal score, typically defined as a dispersion measure of the contribution time series:

$$S_{ij} \;=\; \text{sd}\!\left(\{\text{contrib}_{ij,t}\}_{t=1}^T\right),$$

or equivalently its variance. By the law of total variance, the scalar score decomposes as:

$$
\begin{aligned}
S_{ij}^2 \;=\; & \text{Var}(\text{contrib}_{ij,t}) \\
=\; & \mathbb{E}\!\left[\text{Var}\!\left(\text{contrib}_{ij,t} \mid X_{t-1}^{(i)}\right)\right] \;+\; \text{Var}\!\left(g_{ij}\!\left(X_{t-1}^{(i)}\right)\right),
\end{aligned}
$$

where $g_{ij}(x) := \mathbb{E}[\text{contrib}_{ij,t} \mid X_{t-1}^{(i)} = x]$. Since the first term is non-negative, this gives the inequality

$$S_{ij}^2 \;\geq\; \text{Var}\!\left(g_{ij}\!\left(X_{t-1}^{(i)}\right)\right),$$

with equality if and only if $\text{Var}(\text{contrib}_{ij,t} \mid X_{t-1}^{(i)}) = 0$ almost surely. The scalar score therefore *upper-bounds* the

variance of the function-valued influence, with the gap equal to the expected within-state residual variance. This makes the information bottleneck more severe than a simple projection: $S_{ij}^2$ conflates between-state variation in $g_{ij}(\cdot)$ with within-state residual noise, further obscuring the functional structure of causal influence. Different functions $g_{ij}(\cdot)$ can therefore induce identical or similar scalar scores despite exhibiting fundamentally different behavior across states.

### 3.1. Why Scalar Scores Are a Bottleneck

Scalar scores summarize function-valued influence using a single number; therefore, they inevitably discard information. We highlight three generic failure modes that arise in nonlinear autoregressive models.

**Failure mode 1: Distinct functional forms with similar aggregates**. Qualitatively different causal mechanisms, such as monotonic linear effects, thresholded activation, saturation, or sign-changing behavior, can produce similar dispersion when averaged over the empirical distribution of the source variable. This follows from the non-injectivity of moment-based summaries: distinct functions can share identical low-order moments, and even full moment sequences need not uniquely determine a distribution (Stoyanov, 2013), the statistical analogue of Anscombe's quartet (Anscombe, 1973). As a result, edges governed by very different functions $g_{ij}(\cdot)$ may receive comparable scalar scores $S_{ij}$, making them indistinguishable under score-based ranking or thresholding.

**Failure mode 2: Collapsing regime-dependent influence**. In many systems, causal influence is active only within specific regimes (e.g., low vs. high values of a state variable). Scalar summaries average over these regimes, obscuring where and when an influence operates. This limitation follows from the fact that marginal averaging under mixture distributions discards regime membership information (Friedman, 2001); ICE plots were introduced precisely to recover individual-level heterogeneity that partial dependence obscures (Goldstein et al., 2015). A relationship that is strong but localized to a narrow region of the state space may receive the same score as one that is weak but globally active, despite having very different substantive interpretations.

**Failure mode 3: Interaction-driven effects**. When contributions depend on interactions between variables or on the joint configuration of the system, marginal scalar summaries fail to capture these dependencies. Averaging over other variables can obscure or distort non-additive dependencies (Apley & Zhu, 2020), and conditioning on additional variables can reveal sharp changes in functional form that any single scalar statistic necessarily collapses.

**Conceptual overview** Figure 1 provides a schematic

overview of this representational gap. A trained additive autoregressive model produces a time-indexed set of source-to-target contribution (a contribution tensor) encoding source-specific effects over time. Standard practice reduces this object to a single scalar causal score matrix by aggregating contributions across time. This scalar reduction is convenient for ranking edges, but it collapses the higher-dimensional object learned by nonlinear models - namely, state-dependent causal influence functions. In contrast, our goal is to retain and analyze these function-valued objects by estimating causal response functions $g_{ij}(\cdot)$ (and their regime-conditional extensions) directly from the trained model. Doing so preserves state- and regime-dependent causal behavior. The remainder of the paper demonstrates, through controlled synthetic systems and real panel data, that this difference is not merely conceptual but has substantial practical consequences.

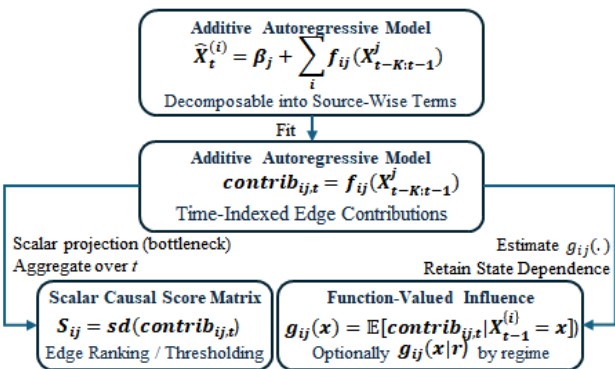

*Figure 1.* From learned contributions to scalar scores versus function-valued causal influence. Additive autoregressive models produce time-indexed contributions $contrib_{ij,t}$ for each directed edge. Standard practice aggregates these into a scalar score $S_{ij}$, enabling edge ranking but discarding state dependence. Function-valued causal influence instead represents each edge as a response function $g_{ij}(x)$, preserving thresholds, saturation, asymmetry, and sign changes that scalar summaries cannot capture.

## 4. Estimating Function-Valued Influence

In our work, we show that a population-level object derived from contribution-decomposable autoregressive models can be estimated directly from a trained model using an intervention-style procedure. Our goal is to demonstrate that function-valued causal influence is operational and computationally tractable, not merely conceptual.

### 4.1. Individual Conditional Expectation as an Intervention-Style Estimator

We estimate function-valued causal influence using **Individual Conditional Expectation (ICE)** (Goldstein et al.,

2015), a model-based procedure that evaluates how a trained predictor's output changes when a specific input variable is intervened upon while holding all other inputs fixed. Consider a trained additive autoregressive model and a fixed target variable $j$. Let

$$\hat{X}_t^{(j)}(\mathbf{Z})$$

denote the model's prediction of $\hat{X}_t^{(j)}$ given lagged inputs $\mathbf{Z} = X_{t-K:t-1}$. For a source variable $i$, ICE proceeds by replacing the lagged values of $X^{(i)}$ with a fixed value $x$, recomputing the prediction, and measuring the change relative to a baseline. Formally, define the ICE effect at value $x$ as

$$\Delta_{ij,t}(x) = \hat{X}_t^{(j)}\Big(X_{t-K:t-1}^{(i)} \leftarrow x\Big) - \hat{X}_t^{(j)}(X_{t-K:t-1}),$$

where $X_{(t-K:t-1)}^{(i)} \leftarrow x$ denotes replacing the lagged inputs of variable i by the constant x while leaving all other variables unchanged. Averaging over time windows yields an estimator of the function-valued causal influence:

$$\hat{g}_{ij}(x) = \mathbb{E}_t[\Delta_{ij,t}(x)].$$

This procedure directly estimates the causal object defined in Section 3.1 by probing the trained model under controlled interventions on the source variable.

## 4.2. Lag-Aggregated ICE

In multivariate time-series models, each source variable appears through multiple lags. To estimate variable-level causal influence rather than lag-specific effects, we use lag-aggregated ICE, in which all lags of the source variable are intervened on simultaneously. Specifically, for a fixed value x, we replace

$$x_{t-1}^{(i)} = x_{t-2}^{(i)} = ... = x_{t-K}^{(i)} = x,$$

and compute the resulting change in the prediction of $X_t^{(j)}$. The lag-aggregated ICE estimator is therefore

$$\hat{g}_{ij}^{\text{agg}}(x) = \mathbb{E}_t\Big[\hat{X}_t^{(j)}\Big(X_{t-1:t-K}^{(i)} = x\Big) - \hat{X}_t^{(j)}(X_{t-K:t-1})\Big].$$

Lag-aggregated ICE provides a single, interpretable response function for each directed edge $i \rightarrow j$, aligning with the variable-level causal influence defined in Section 3. We note that this estimator targets a *sustained-value intervention* - setting all lags of variable $i$ to the constant $x$, which is a distinct population object from the single-lag conditional expectation $g_{ij}(x) = \mathbb{E}[\text{contrib}_{ij,t} \mid X_{t-1}^{(i)} = x]$ defined in Section 3. This distinction is a deliberate design choice: lag-aggregated ICE provides a variable-level response function that is more interpretable and more policy-relevant

than a lag-specific conditional, at the cost of targeting a different (though related) causal object. Under stationarity, when contributions are primarily driven by the most recent lag with earlier lags showing attenuated magnitude due to temporal decay, the two objects are approximately aligned. Additional details on lag-specific ICE analysis, including empirical results, are presented in Appendix C.

## 4.3. Regime-Conditional ICE

Many nonlinear systems exhibit regime-dependent causal behavior, in which the influence of a source variable differs across states of the system. To capture such heterogeneity, we estimate regime-conditional ICE curves. Let $r_t$ denote a regime variable, such as a discretized index or the lagged value of the target variable. Conditioning ICE on $r_t$ yields

$$\hat{g}_{ij}(x \mid r) = \mathbb{E}_t[\Delta_{ij,t}(x) \mid r_t = r].$$

In practice, we estimate this quantity by partitioning time windows into discrete regime bins (e.g., low, medium, and high values of $r_t$) and computing ICE curves separately within each bin. This reveals threshold effects, asymmetries, and state-specific activation patterns that are averaged out in unconditional summaries.

## 4.4. Practical Considerations

For panel time series, we compute ICE over lag windows that respect unit boundaries. Interventions are applied within each unit's history, and expectations are taken over all valid windows across units.

**Normalization.** Because ICE operates on model inputs, the scale of interventions must match the scale used during training. When models internally normalize inputs, ICE interventions are applied in the normalized space to ensure consistency.

**Computational Cost.** ICE requires recomputing predictions for each intervention value and time window. In practice, this cost is manageable via batching and GPU acceleration, and scales linearly with the number of grid points used to evaluate $x$.

**Interpretation** We emphasize that ICE, as used here, is not a post-hoc explanation of individual predictions. Instead, it is an estimator for a well-defined causal object: the function-valued causal influence $g_{ij}(\cdot)$, derived from the additive structure of the trained model. Unlike local explanation methods that attribute importance to individual features for specific instances, ICE in our framework serves as a global, intervention-style operator that recovers how a learned causal relationship behaves across the state space. This distinction is crucial for interpreting nonlinear causal models and for connecting their outputs to theory-driven

reasoning in applied domains.

**Out-of-distribution inputs.** A known limitation of ICE and partial dependence methods is that constant substitutions may generate input combinations not observed during training, potentially biasing estimated curves in regions of sparse data coverage (Apley & Zhu, 2020). In our synthetic experiments, the jump process ($p = 0.15$, amplitude $1.5\sigma$) is designed to ensure adequate coverage of nonlinear regimes; ICE evaluations are confined to the empirical range of the source variable. In the democracy application, interventions are applied within the standardized empirical range of each variable across all country-year observations. Users should interpret ICE curves with caution in extrapolation regions where training data coverage is sparse. Pointwise bootstrap confidence bands for all four democracy edges are reported in Appendix G.

In the following section, we apply this estimation framework to controlled synthetic systems and to a real panel dataset on democratic development. These experiments demonstrate that ICE-based function-valued analysis recovers meaningful causal structure that is systematically invisible to scalar causal scores.

## 5. Experiments

### 5.1. Synthetic Systems

**Experimental setup**. We construct a controlled synthetic system with three variables $X_t, Y_t, Z_t$, where only $X$ causally influences $Y$ through a known nonlinear mechanism. The source variable $X_t$ follows a stationary autoregressive process with occasional jumps to ensure coverage of nonlinear regimes, while $Y_t$ depends on $X_{t-1}$ through a known function $g(\cdot)$. The target variable $Y$ depends on the lagged value of $X$ through one of four distinct causal mechanisms - linear, thresholded, saturating, or sign-changing, designed to have comparable overall strength but different functional form. All other relationships are purely autoregressive. This design allows us to isolate the representational properties of learned causal influence while holding overall signal strength and noise constant across systems.

The data-generating process is:

$$X_t = 0.6\,X_{t-1} + \varepsilon_t^X,$$
$$Y_t = 0.3\,Y_{t-1} + g(X_{t-1}) + \varepsilon_t^Y,$$
$$Z_t = 0.6\,Z_{t-1} + \varepsilon_t^Z,$$

where $\varepsilon_t^X, \varepsilon_t^Y, \varepsilon_t^Z \sim \mathcal{N}(0, \sigma^2)$ are independent Gaussian noise terms. A complete specification of the synthetic system, along with robustness checks demonstrating that our results do not depend on particular simulation parameters, is provided in Appendix B.

**Scalar causal scores.** Despite substantial differences in the underlying causal response functions - linear, thresholded, saturating, and sign-changing - the resulting avrage scalar scores are tightly clustered (Table 1). Across 15 runs per system, mean scores range from 0.63 to 0.65, with overlapping minima and maxima and comparable variability across seeds.

This overlap implies that scalar causal scores provide little information about the functional form of the causal relationship. In particular, sign-changing and threshold mechanisms receive scalar scores nearly indistinguishable from those of linear and saturating mechanisms, despite exhibiting fundamentally different behavior across the state space. Consequently, ranking or thresholding edges based on scalar scores alone cannot distinguish among qualitatively different causal dynamics. Robustness of scalar causal scores is further discussed in Appendix D.

*Table 1.* Summary of NAVAR scalar causal scores for the edge $X \rightarrow Y$ across synthetic systems.

| SYSTEM | RUNS | MEAN | STD. | MIN | MAX |
|---|---|---|---|---|---|
| LINEAR | 15 | 0.626 | 0.015 | 0.601 | 0.664 |
| SATURATING | 15 | 0.650 | 0.011 | 0.630 | 0.675 |
| SIGN-CHANGING | 15 | 0.634 | 0.017 | 0.610 | 0.670 |
| THRESHOLD | 15 | 0.632 | 0.015 | 0.605 | 0.668 |

**Function-valued causal influence.** Results shown in Table 1 illustrate a core representational limitation: scalar summaries collapse function-valued causal influence into a single magnitude that averages over regimes. In contrast, as shown in Figure 2, the corresponding estimated causal response functions reveal substantial qualitative differences. The linear mechanism exhibits a monotonic, approximately linear response. The threshold mechanism displays a flat region around zero followed by sharp activation, consistent with a regime-switching effect. The saturating mechanism shows diminishing marginal influence at large magnitudes of the source variable, while the sign-changing mechanism exhibits non-monotonic behavior with a reversal in slope across regimes. Notably, these differences are recovered despite the similarity of scalar causal scores across systems.

**Interpretation** This experiment demonstrates that scalar causal scores constitute a severe information bottleneck in nonlinear time-series models. While they may capture the overall strength or variability of an influence and may be sufficient for detecting the presence of a causal relationship, they discard essential functional structure - precisely the structure that governs how causal effects manifest across states of the system. As a result, two edges with indistinguishable scalar scores can encode radically different causal dynamics. Function-valued analysis recovers this structure directly, providing a more faithful representation of the causal objects learned by nonlinear autoregressive models. Quantitative recovery metrics (Pearson correlations

between the true $g(\cdot)$ and estimated ICE curves, averaged across 15 runs) are reported in Appendix H.

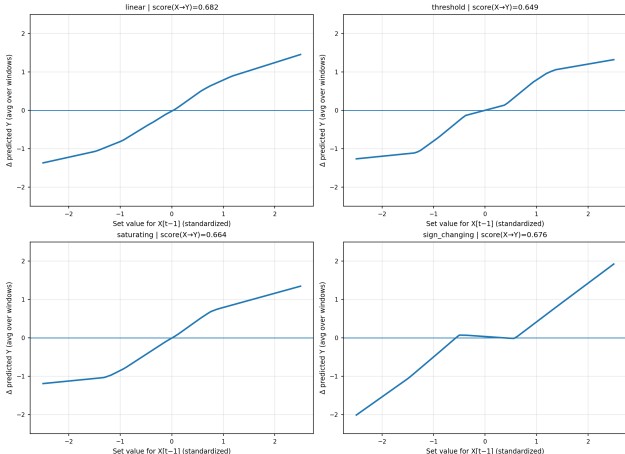

*Figure 2.* Same scalar score, different causal functions. Despite matched scalar scores, the recovered functions exhibit qualitatively distinct behaviors: smooth linear dependence, abrupt threshold activation, saturation at extreme values, and sign reversal across regimes. These differences correspond to fundamentally different causal mechanisms that cannot be inferred from a single scalar summary.

## 5.2. Democratic Development (Function-Valued Causal Influence in a Real Panel Time Series)

The purpose of this experiment is to evaluate whether the representational limitations identified in Section 5.1 are consequential in a realistic, theory-driven setting. Specifically, the democracy experiments serve as (i) a stress test for function-valued causal influence in the presence of persistence, noise, and cross-dependence; (ii) a demonstration that scalar causal scores can obscure regime-specific behavior in applied settings; and (iii) an application where interpretation of causal mechanisms is substantively important. This section is not intended as a benchmark comparison, a forecasting competition, a structural causal analysis of democracy, or a significance-testing exercise. Those questions are addressed elsewhere. Here, the focus is exclusively on representation and interpretation: what nonlinear causal models learn, how those learned relationships are summarized, and what information is lost when summaries are reduced to scalars.

### 5.2.1. DATA SETUP

**Dataset Characteristics.** We analyze a panel of 139 countries observed annually over a period of 35 years. The source of data for this experiment is the **Varieties of Democracy (V-Dem)** project (Coppedge et al., 2025; Pemstein et al.,

2025). Details on the democracy dataset, preprocessing steps, and panel construction are provided in Appendix E.

The dataset for the study comprises 16 democracy-related indicators capturing distinct institutional dimensions, including freedom of association, freedom of expression, electoral integrity, judicial and legislative constraints on the executive, civil society participation, and equality before the law. Each country contributes a relatively short time series (35 years), and the variables exhibit strong temporal persistence as well as substantial cross-correlation, reaching, in some cases, the magnitude of over $0.9$ (as described in Appendix E). These properties reflect the complexity of institutional change and pose challenges for nonlinear causal discovery.

**Modeling choice.** We estimate a nonlinear additive autoregressive model using NAVAR (Bussmann et al., 2021) in panel mode, with fixed-length segments corresponding to each country's time series. We allow for multiple lags (up to $K = 8$) to capture medium-term institutional dynamics and use identical hyperparameters across all analyses. This setup contrasts deliberately with the synthetic experiments in Section 5.1, which use a clean, lag-1 system with minimal persistence. The comparison highlights how representational issues manifest differently in idealized synthetic versus realistic, multi-lag settings.

### 5.2.2. SCALAR CAUSAL GRAPH

As a baseline, we compute the NAVAR scalar causal score matrix and visualize a subset of key variables. This scalar graph corresponds to standard practice in nonlinear causal discovery: edges are summarized by a single magnitude, optionally thresholded or normalized, and interpreted as a network of mutual influence. The causal graph matrix is presented in Appendix E.

The resulting graph appears dense and intuitively plausible, with many democracy components exhibiting bidirectional or mutually reinforcing relationships, supporting findings of previous research (Coppedge et al., 2020; 2022). However, many edges have similar scalar magnitudes, making it difficult to distinguish among them or to infer how these relationships operate. The scalar graph suggests a web of interdependence, but provides no insight into regime dependence, asymmetry, or nonlinear activation across levels of democratic development.

### 5.2.3. FUNCTION-VALUED ANALYSIS VIA ICE

We next apply the same ICE-based intervention framework used in the synthetic study to the democracy panel.

**Selection of Edges.** We focus on a small number of substantively important edges that (i) have non-trivial scalar scores, (ii) are theoretically meaningful, and (iii) admit different interpretations. We select "Free and Fair Elections"

("clean_elections" in the dataset) as it has been shown in classical political science literature as one of the most important indicators of democracy (Dahl, 1966; 2000). Examples include relationships between this variable and four other extensively studied variables: freedom of expression, freedom of association, judicial constraints, and legislative constraints. We deliberately limit the analysis to four edges; a small number of well-chosen cases is more informative than exhaustive enumeration. A broader quantitative survey of functional heterogeneity across all 50 edges above the score threshold is provided in Appendix I.

**ICE Response Curves** For each selected edge $X \to Y$, we construct lag windows from the panel data and intervene on the source variable (using a lag-aggregated intervention) while holding all other inputs fixed. We compute

$$\Delta Y_t(x) = \mathbb{E}[\hat{Y}_t | X_{t-1} = x] - \mathbb{E}[\hat{Y}_t],$$

and plot the resulting response functions with the source variable standardized on the x-axis and the average change in the predicted target on the y-axis.

Across edges, the ICE curves reveal patterns that are invisible in the scalar graph, including threshold effects (e.g., institutional changes matter only above a minimum level), asymmetry between improvements and declines, saturation at high levels, and, in some cases, non-monotonic behavior.

### 5.2.4. REGIME-SPECIFIC INTERPRETATION

The democracy setting allows us to interpret these response functions in terms of developmental regimes (low, medium, and high levels of democracy as captured by the "clean_elections." Figure 3 shows lag-aggregated ICE curves for four substantively important edges targeting clean elections. Although the corresponding scalar causal scores are small and of similar magnitude ($\approx 0.1$ for "judicial constraints" and "freedom of association" variables, and $\approx 0.2$ for "freedom of expression" and "legislative constraints"), suggesting weak and roughly uniform influence, the function-valued responses reveal substantial heterogeneity across levels of the source variables and across developmental regimes.

For freedom of expression and judicial constraints, the estimated effects are essentially flat in low regimes and become strongly positive only after a moderate institutional threshold is crossed, indicating that these freedoms matter primarily beyond a minimum baseline. Freedom of expression exhibit a smoother, more gradual activation pattern, while legislative constraints display strong asymmetry: improvements from low to moderate levels are associated with sizable gains in predicted electoral quality, followed by clear saturation at higher levels. These qualitative differences are not reflected in scalar scores, which treat the edges as

broadly comparable.

The results show that nearly identical scalar causal summaries obscure threshold effects, asymmetry, and saturation that are central to interpreting democratic development. Function-valued analysis recovers these regime-specific patterns directly, demonstrating that the representational collapse identified in Section 5.1 has concrete consequences in a real, theory-driven panel dataset. This mirrors the findings of the synthetic experiments (Section 5.1) but shows that the representational collapse of scalar summaries has concrete consequences in a real, theory-driven dataset.

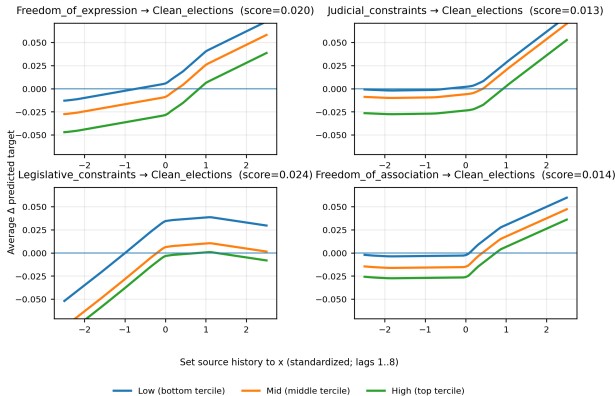

*Figure 3.* Each panel shows lag-aggregated ICE curves for the effect of a source variable on target ("clean elections"), holding all other inputs fixed. Curves are split by tertiles of the target's lagged value (low, mid, high regimes), revealing strong regime dependence. Although the corresponding scalar causal scores are small and of similar magnitude, the functional forms differ substantially across edges, including threshold activation, asymmetric effects, and regime-specific sensitivity. Scalar summaries therefore fail to capture how causal influence operates in realistic panel time-series systems.

## 6. Discussion, Implications, and Limitations

This paper reframes causal discovery in nonlinear time series as a problem of representation. Nonlinear autoregressive models learn rich, state-dependent causal structure, yet standard practice summarizes this structure using scalar causal scores. Our results show that this scalar aggregation discards essential information about how causal influence operates across states of the system.

### 6.1. Implications

**Causal discovery and evaluation.** Scalar causal matrices remain useful for coarse screening and visualization, but they are insufficient as primary summaries of nonlinear causal structure. Function-valued causal influence provides

a more faithful representation by retaining thresholds, saturation, asymmetry, and sign changes. This reframing also suggests new evaluation directions: beyond edge detection metrics, models can be compared based on whether they recover qualitatively correct causal response functions.

**Interpretability.** Function-valued analysis offers interpretability that is intrinsic to the model rather than post hoc. In the democracy application, scalar scores suggest a dense network of weak, similar influences, whereas function-valued analysis reveals regime-specific mechanisms that activate only after institutional thresholds are crossed or that saturate at high levels. These distinctions are essential for substantive interpretation but are systematically obscured by scalar summaries.

**Applied and policy-facing use.** Scalar causal scores are often interpreted as measures of effect strength or priority in applied settings. Our findings caution against such use. Scalar summaries can mask regime dependence and nonlinearity, leading to misleading conclusions about when and how interventions matter. Function-valued analysis makes these contingencies explicit, reducing the risk of misinterpretation without requiring new model architectures.

**Downstream applications.** Function-valued causal influence supports a broader inference pipeline beyond causal discovery. Formal forecast-based edge ablation via Diebold-Mariano testing complements scalar scores by identifying which discovered edges are genuinely necessary for prediction - a distinction that scalar summaries alone cannot make (Kuskova et al., 2026). NAVAR-derived causal structures, enriched by function-valued analysis, can serve as structural priors for time-varying network autoregression supporting impulse response and counterfactual analysis in panel time series (Zaytsev et al., 2026a). Applied to trajectory-aware reliability modeling across democratic systems, these dynamic causal structures can predict institutional failure risk from degradation propagation paths better than traditional models (Zaytsev et al., 2026b).

### 6.2. Limitations

Several limitations should be noted. First, estimating causal response functions depends on data coverage; ICE-based estimates may be noisy in sparsely observed regions of the state space. Second, our analysis relies on additive, contribution-decomposable models; the additivity assumption ensures source-variable separability, which is what enables ICE to isolate individual causal response functions cleanly. Models with explicit cross-variable interactions would require richer decomposition methods, such as Shapley interaction indices, to disentangle joint effects. In such models, interaction effects, if present, would appear as residual heterogeneity in regime-conditional curves rather than being attributable to specific variable pairs. Extending

function-valued causal analysis to non-additive architectures is an important direction for future work. Finally, causal interpretation remains predictive and Granger-style: the response functions characterize model behavior rather than structural or interventional causal effects.

### 6.3. Closing perspective

Overall, our results suggest that the opacity often attributed to nonlinear causal models arises less from the models themselves than from how their outputs are summarized. By retaining the function-valued structure that these models already learn, researchers can recover interpretable, regime-specific causal mechanisms while preserving the flexibility of modern nonlinear approaches. Code and data for this study are available on GitHub at `https://github.com/vkuskova/ICML2026-28194`.

## Impact Statement

This work contributes to the interpretability of nonlinear causal discovery methods by clarifying how learned causal structure is represented and summarized. By reframing causal influence as a function-valued object, our approach enables practitioners to better understand when and how causal relationships operate across different states of a system. This can benefit researchers in domains that rely on theory-driven causal reasoning, such as the social sciences, economics, and public policy, where opaque scalar summaries may hinder adoption or lead to misinterpretation of model outputs.

## Acknowledgements

This research is made possible in part by support from the Franco Institute for Liberal Arts and the Public Good, College of Arts & Letters; the Kellogg Institute for International Studies; and the Lucy Family Institute for Data & Society, University of Notre Dame.

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

# Appendix

## A. Neural Additive Vector Autoregression (NAVAR) Implementation

Neural Additive Vector Autoregression (NAVAR) (Bussmann et al., 2021) is a flexible nonlinear extension of classical vector autoregression that preserves interpretability through an additive structure. In NAVAR, each target variable is modeled as a sum of source-specific nonlinear functions applied to lagged inputs, allowing causal influence to be decomposed at the level of directed variable pairs.

### A.1. Model Architecture

For a multivariate time series $\{X_t\}_{t=1}^{T}$, NAVAR models each target variable $X_t^{(j)}$ as

$$\hat{X}_t^{(j)} = \beta_j + \sum_{i=1}^{N} f_{ij}\left(X_{t-K:t-1}^{(i)}\right),$$

where $K$ is the maximum lag length and each function $f_{ij}(\cdot)$ is parameterized by a small feedforward neural network. Each $f_{ij}$ receives the lagged history of a single source variable $i$ as input and produces a scalar contribution to the prediction of target variable $j$. Additivity is enforced by construction, ensuring that contributions from different source variables are separable.

In our implementation, each $f_{ij}$ consists of a fully connected network with one hidden layer and ReLU nonlinearities (Laurent & Brecht, 2018), followed by a linear output layer. The number of hidden units is held fixed across all source–target pairs. This architecture balances expressive power with stability and interpretability, and it allows learned contributions to be directly inspected and aggregated.

### A.2. Training Procedure

NAVAR models are trained by minimizing a standard predictive loss (mean squared error) between observed values and one-step-ahead predictions. Training is performed using stochastic gradient descent with the Adam optimizer (Zhang, 2018). To reduce overfitting and encourage stable contribution functions, we apply both dropout and weight decay during training. An $l_1$ regularization penalty is applied to the outputs of the contribution networks to promote sparsity in the learned causal graph.

For panel data, the time series for each unit is segmented into fixed-length sequences, and training windows are constructed so that lag windows do not cross unit boundaries. All experiments use identical hyperparameters across systems unless otherwise noted, ensuring that differences in learned behavior arise from differences in the data-generating process rather than from model tuning.

### A.3. Causal Scores and Contribution Tensors

After training, NAVAR produces a contribution tensor $\{contrib_{ij,t}\}$, where

$$S_{i,j} = sd(\{contrib_{ij,t}\}_{t-1}^{T})$$

These scalar scores are used only for comparison with function-valued causal influence and are not the primary object of analysis in this paper.

*Table 2.* NAVAR hyperparameters used in all experiments.

| Hyperparameter | Value |
|---|---|
| Maximum lag length ($K$) | 8 |
| Hidden layers per $f_{ij}$ | 1 |
| Hidden units per layer | 32 |
| Activation function | ReLU |
| Output activation | Linear |
| Dropout rate | 0.10 |
| Weight decay ($\ell_2$) | $3 \times 10^{-4}$ |
| Sparsity penalty ($\ell_1$) | 0.15 |
| Optimizer | Adam |
| Learning rate | $3 \times 10^{-4}$ |
| Batch size | 128 |
| Training epochs | 1000 |
| Validation split | 10% |
| Input normalization | Yes (z-score) |
| Panel segment length | 35 (democracy data) |
| Recurrent components | None (feedforward only) |
| Random seed | Fixed per run |

### A.4. Relationship to function-valued analysis

Crucially, our proposed function-valued causal influence does not modify the NAVAR architecture or training procedure. Instead, it operates on the learned contribution tensor produced by NAVAR, treating the source-specific functions $f_{ij}(\cdot)$ as estimable causal objects. As a result, the methods introduced in this paper are directly applicable to any additive, contribution-decomposable autoregressive model, including but not limited to NAVAR.

### A.5. Reproducibility

### A.6. Reproducibility

All experiments are run with fixed random seeds controlling NumPy and PyTorch randomness. Hyperparameter values used in each experiment are reported alongside the corresponding results. Code and data to reproduce all experiments and figures are available at https://github.com/vkuskova/ICML2026-28194. Unless otherwise noted, the same configuration is used for all synthetic and empirical experiments to ensure comparability across systems.

## B. Synthetic Data Generation Details.

This section provides full details of the synthetic data–generating process used in Section 5.1. The goal of the synthetic experiments is to isolate representational properties of nonlinear causal models under controlled conditions, ensuring that differences in learned behavior arise from functional form rather than trivial differences in noise, persistence, or scale.

### B.1. Data Generating Process

We consider a three-variable system $(X_t, Y_t, Z_t)$ observed over $t = 1, \ldots, T$, where:

- $X_t$ is the source variable,

- $Y_t$ is the target variable, and

- $Z_t$ is a control (distractor) variable.

Only the directed edge $X \to Y$ is causal; all other dependencies are autoregressive. The system evolves according to:

$$X_t = 0.6\, X_{t-1} + \varepsilon_t^X + J_t,$$
$$Y_t = 0.3\, Y_{t-1} + g(X_{t-1}) + \varepsilon_t^Y,$$
$$Z_t = 0.6\, Z_{t-1} + \varepsilon_t^Z,$$

where $\varepsilon_t^X, \varepsilon_t^Y, \varepsilon_t^Z \sim \mathcal{N}(0, \sigma^2)$ are independent Gaussian noise terms.

### B.2. Jump Process for Source Variable

To ensure that nonlinear causal mechanisms are sufficiently expressed, the source variable $X_t$ includes an occasional jump component:

$$J_t = \begin{cases} +1.5, & \text{with probability } p/2, \\ -1.5, & \text{with probability } p/2, \\ 0, & \text{with probability } 1 - p, \end{cases}$$

with p=0.15. This jump process forces the system to visit regions of the state space where nonlinear effects (e.g., thresholds and saturation) activate. Without this component, the autoregressive dynamics would concentrate $X_t$ near zero, causing nonlinear mechanisms to be effectively linearized by the learning algorithm. All causal mechanisms are exposed to the same jump process to ensure comparability.

### B.3. Standardization

After simulation, all variables are standardized column-wise:

$$\tilde{X}_t^{(k)} = \frac{X_t^{(k)} - \mu_k}{\sigma_k},$$

where $\mu_k$ and $\sigma_k$ denote the empirical mean and standard deviation of the variable $k \in X, Y, Z$. Standardization ensures numerical stability during training and places all variables on a comparable scale.

### B.4. Causal Mechanisms

We consider four qualitatively distinct causal mechanisms $g(\cdot)$ governing the edge $X \to Y$. Each mechanism is designed to have comparable overall strength while differing in functional form.

**Linear Mechanism.** Linear mechanism is defined as

$$g_{\text{linear}}(x) = \text{clip}(x, -1.2,\ 1.2).$$

It produces a monotonic, approximately linear response over the central region of the state space, with bounded tails to prevent rare extreme values from dominating variability.

**Threshold mechanism.** Defined as

$$g_{\text{threshold}}(x) = \begin{cases} a\ \text{sign}(x), & |x| > c, \\ 0, & |x| \leq c, \end{cases}$$

with threshold $c = 0.6$ and amplitude $a = 1.6$, it represents a regime-switching effect in which causal influence activates only once the source variable exceeds a critical magnitude.

**Saturating mechanism.** Defined as

$$g_{\text{saturating}}(x) = \text{clip}(x, -1.0,\ 1.0),$$

this mechanism exhibits diminishing marginal influence at large magnitudes of the source variable, capturing saturation effects common in nonlinear systems.

**Sign-changing mechanism** is defined as

$$g_{\text{sign-changing}}(x) = \begin{cases} -x, & |x| < c, \\ x, & |x| \geq c, \end{cases}$$

with c=0.6. In this case, the source variable has a negative effect near the origin and a positive effect in the tails, producing a non-monotonic causal response.

### B.5. Design Rationale

All four mechanisms:

- operate on the same autoregressive backbone,

- are subject to identical noise and jump processes,

- are standardized to comparable scale, and

- are learned using identical model hyperparameters.

As a result, differences in learned causal behavior arise from functional form rather than differences in effect magnitude, persistence, or noise. This design allows us to isolate the representational consequences of scalar causal summaries and to demonstrate how qualitatively distinct causal mechanisms can collapse to similar scalar scores despite exhibiting fundamentally different response functions.

### B.6. Algorithm for Synthetic Data Generation

Algorithm 1 provides pseudocode for generating the synthetic time series used in Section 5.1. The algorithm ensures that nonlinear causal mechanisms are sufficiently expressed while holding persistence, noise, and scale constant across systems.

### B.7. Robustness to Jump Probability and Noise Level

To verify that our results do not depend on a finely tuned simulation design, we evaluate robustness across different jump probabilities and noise levels. For each configuration, we recompute scalar causal scores for the edge $X \rightarrow Y$ under all four causal mechanisms. Table 1 of the main body reports representative scalar scores (mean across seeds) under varying parameters. While absolute magnitudes vary modestly, the central qualitative result - similar scalar scores across qualitatively different mechanisms - remains unchanged.

---

**Algorithm 1** Synthetic data generation with controlled nonlinear causal mechanisms

---

**Require:** Time length $T$; noise variance $\sigma^2$; jump probability $p$; jump magnitude $\delta$; causal function $g(\cdot)$.
**Ensure:** Standardized time series $\{(X_t, Y_t, Z_t)\}_{t=1}^{T}$.
 1: Initialize $X_1 \leftarrow 0$, $Y_1 \leftarrow 0$, $Z_1 \leftarrow 0$.
 2: **for** $t = 2$ TO $T$ **do**
 3:    Draw noise terms $\varepsilon_t^X, \varepsilon_t^Y, \varepsilon_t^Z \sim \mathcal{N}(0, \sigma^2)$.
 4:    Draw jump variable

$$
J_t = \begin{cases} +\delta, & \text{with probability } p/2, \\ -\delta, & \text{with probability } p/2, \\ 0, & \text{with probability } 1 - p. \end{cases}
$$

 5:    Update source variable: $X_t \leftarrow 0.6\, X_{t-1} + \varepsilon_t^X + J_t$.
 6:    Update target variable: $Y_t \leftarrow 0.3\, Y_{t-1} + g(X_{t-1}) + \varepsilon_t^Y$.
 7:    Update control variable: $Z_t \leftarrow 0.6\, Z_{t-1} + \varepsilon_t^Z$.
 8: **end for**
 9: Standardize each variable to zero mean and unit variance.
10: **Return** $\{(X_t, Y_t, Z_t)\}_{t=1}^{T}$.

---

**Interpretation.** Across all settings, scalar scores remain within a narrow band and do not reliably distinguish among linear, thresholded, saturating, and sign-changing mechanisms. This confirms that the representational collapse observed in Section 5.1 is not an artifact of a particular choice of simulation parameters.

## C. Lag-Specific ICE Analysis

This section reports lag-specific Individual Conditional Expectation (ICE) results, which decompose function-valued causal influence by individual lag positions. While the main text focuses on lag-aggregated ICE for variable-level interpretation, lag-specific analysis serves as a diagnostic to verify that the aggregated response functions are not artifacts of a single lag.

Recall the additive autoregressive model

$$
\hat{X}_t^{(j)} = \beta_j + \sum_{i=1}^{N} f_{ij}\left( X_{t-K:t-1}^{(i)} \right).
$$

Lag-specific ICE isolates the effect of a single lag $l \in \{1, \ldots, K\}$ of a source variable $i$. Let $X_{t-l}^{(i)}$ denote the $l - th$ lag (with $l = 1$ corresponding to $t - 1$). For a trained model $\hat{f}$ define the lag-specific intervention at value $x$ as replacing only that lag:

$$
X_{t-\ell}^{(i)} \leftarrow x,
$$

while keeping all other inputs unchanged. The lag-specific ICE effect at time $t$ is

$$
\Delta_{ij,t}^{(\ell)}(x) = \hat{X}_t^{(j)}\left( X_{t-\ell}^{(i)} \leftarrow x \right) - \hat{X}_t^{(j)}(X_{t-K:t-1}).
$$

Averaging over valid time windows yields the lag-specific response function

$$
\hat{g}_{ij}^{(\ell)}(x) = \mathbb{E}_t\left[ \Delta_{ij,t}^{(\ell)}(x) \right].
$$

This object characterizes how the causal influence of $i \to j$ varies as a function of the $l$-th lag alone.

## C.1. Relationship to Lag-Aggregated ICE

The lag-aggregated ICE used in the main text intervenes simultaneously on all lags of the source variable:

$$X_{t-1}^{(i)} = X_{t-2}^{(i)} = \cdots = X_{t-K}^{(i)} = x,$$

and estimates

$$\hat{g}_{ij}^{\mathrm{agg}}(x) = \mathbb{E}_t\Big[ \hat{X}_t^{(j)}\Big( X_{t-1:t-K}^{(i)} = x \Big) - \hat{X}_t^{(j)}(X_{t-K:t-1}) \Big].$$

Lag-specific ICE provides a decomposition of this aggregated effect across individual lag positions. In practice, we find that the qualitative shape of $\hat{g}_{ij}^l(x)$ is consistent across lags, with earlier lags typically exhibiting attenuated magnitude due to temporal decay. This consistency provides qualitative support for the interpretation of $\hat{g}_{ij}^{\mathrm{agg}}(x)$ as a stable, variable-level causal response, though it does not constitute a formal equivalence proof between the lag-aggregated and single-lag conditional objects.

## C.2. Regime-Conditional Lag-Specific ICE

Lag-specific ICE can also be conditioned on a regime variable $r_t$. Let $b(t)$ denote the regime bin to which time tbelongs. The regime-conditional lag-specific response is

$$\hat{g}_{ij}^{(\ell)}(x \mid b) = \mathbb{E}_t\Big[ \Delta_{ij,t}^{(\ell)}(x) \;\Big|\; b(t) = b \Big].$$

Conditioning reveals whether temporal localization of causal influence differs across regimes. In our experiments, regime-dependent patterns observed in lag-aggregated ICE are also present at the lag-specific level, indicating that regime heterogeneity is not driven by a single lag.

## C.3. Empirical Findings

Across both synthetic systems and the democracy panel, lag-specific ICE curves exhibit the following patterns:

- Qualitative consistency: The functional form (linear, thresholded, saturating, or sign-changing) is preserved across lag positions.

- Magnitude decay: Effects generally weaken for larger l, reflecting autoregressive persistence and diminishing temporal influence.

- Aggregation validity: No single lag dominates the aggregated response; lag-aggregated ICE reflects a stable combination of lag-specific effects.

To validate the use of lag-aggregated ICE as a principled and interpretable summary of variable-level causal influence, we further complement the theoretical lag-specific ICE analysis with empirical results on the threshold mechanism, providing direct evidence that the lag-aggregated response function is supported by consistent patterns across individual lag positions.

### C.3.1. SETUP

We train NAVAR with $K = 4$ lags on a single realization of the threshold synthetic system ($T = 2000$, noise $\sigma = 0.5$, seed 1000). We then compute both the lag-aggregated ICE curve (all lags set simultaneously to $x$) and four lag-specific ICE curves (each intervening on one lag position at a time while holding others fixed). The true mechanism $g_{\mathrm{threshold}}(x)$ is shown as a reference, scaled and centered to match the ICE delta scale.

C.3.2. RESULTS

Figure 4 shows the lag-aggregated curve alongside each lag-specific curve. Lag 1 (the most recent lag, $t-1$) carries the dominant causal signal, recovering a smooth approximation of the true threshold function with large magnitude. Lags 2 through 4 show progressively attenuated magnitude, with the oldest lag (lag 4, $t-4$) contributing a nearly flat response. This monotonic attenuation is consistent with temporal decay in the autoregressive system: recent observations carry more predictive information about the target than distant observations, and NAVAR's learned contribution functions reflect this structure.

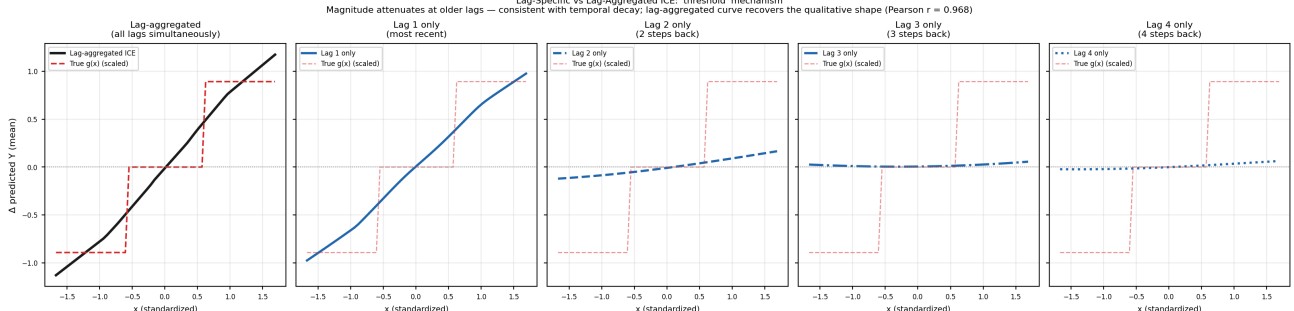

*Figure 4.* Lag-specific vs lag-aggregated ICE for the threshold mechanism ($K = 4$ lags). The leftmost panel shows the lag-aggregated curve (all lags intervened simultaneously), which closely approximates the true threshold function (dashed red). Panels 2–5 show lag-specific curves (one lag at a time): lag 1 (most recent) carries the dominant signal, with magnitude attenuating monotonically at older lags, consistent with temporal decay. The lag-aggregated curve recovers the qualitative shape with high fidelity (Pearson $r = 0.968$).

The lag-aggregated curve, which combines all lags simultaneously, achieves the highest fidelity to the true mechanism (Pearson $r = 0.968$), confirming that aggregation recovers stable variable-level causal structure even when individual lags contribute unequally. These results support the interpretation of lag-aggregated ICE as a reliable summary of function-valued causal influence at the variable level.

C.3.3. RATIONALE FOR FOCUS ON LAG-AGGREGATED ICE

Lag-specific ICE is informative for diagnostic purposes but less suitable for presentation in theory-driven domains. Interpreting multiple lag-specific curves can obscure the central question of how a variable influences another across states and regimes, shifting attention to fine-grained temporal mechanics. Accordingly, we present lag-aggregated ICE in the main text and report lag-specific analyses here to demonstrate robustness and transparency.

## D. Robustness of Scalar Causal Scores

This section presents an evaluation of the robustness of scalar causal scores to random initialization, simulation parameters, and alternative aggregation choices. The goal is to demonstrate that the representational collapse identified in Section 5.1 is systematic and not an artifact of a particular seed, noise realization, or scalar definition. The similarity of scalar scores across mechanisms is not accidental or fragile. Rather, it reflects a systematic property of scalar aggregation in nonlinear autoregressive models. As shown in the main text, function-valued causal influence recovers the missing structure directly, providing a more faithful representation of the causal objects learned by the model.

Scalar causal scores in additive autoregressive models are computed as dispersion measures of time-

indexed contribution series. To assess robustness, we repeat the synthetic experiments in Section 5.1 across multiple random seeds controlling both data generation and model initialization.

For each mechanism $m \in \{$"linear","threshold","saturating","sign-changing"$\}$, and each seed $s$, we compute the scalar score

$$S_{ij}^{(m,s)} = \mathrm{sd}\left(\left\{\mathrm{contrib}_{ij,t}^{(m,s)}\right\}_{t=1}^{T}\right),$$

where $contrib_{ij,t}^{m,s}$ denotes the learned contribution of source variable $i$ to target variable $j$ at time $t$. Across seeds, the scalar scores remain tightly clustered for all mechanisms, with overlapping ranges and consistent rank orderings. While small fluctuations in magnitude occur due to stochastic optimization and noise realizations, no mechanism becomes separable from the others based on scalar score alone. This confirms that the similarity of scalar scores observed in Section 5.1 is not driven by a particular initialization.

### D.1. Robustness to Noise Level and Jump Probability

To verify that scalar score similarity does not depend on a finely tuned simulation regime, we vary key parameters of the synthetic data-generating process:

- noise standard deviation $\sigma \in \{0.2, 0.3, 0.4\}$,
- jump probability $p \in \{0.10, 0.15, 0.20\}$.

For each configuration, we recompute scalar scores using the same trained model architecture and hyperparameters. While absolute magnitudes of scalar scores vary with signal-to-noise ratio, the relative proximity of scores across mechanisms is preserved. In particular, linear, thresholded, saturating, and sign-changing mechanisms continue to occupy a narrow band of scalar values, despite exhibiting qualitatively different response functions. These results indicate that the representational collapse documented in Section 5.1 is robust to moderate changes in the data distribution and does not rely on a specific choice of simulation parameters.

### D.2. Alternative Scalar Aggregation Metrics

One might ask whether the observed collapse is specific to the standard deviation-based scalar score or whether alternative scalar summaries recover additional information. To address this, we consider several alternative scalar metrics computed from the contribution series:

$$S_{ij}^{\mathrm{mean}} = \mathbb{E}_t\left[\left|\mathrm{contrib}_{ij,t}\right|\right],$$
$$S_{ij}^{\mathrm{max}} = \max_t \left|\mathrm{contrib}_{ij,t}\right|,$$
$$S_{ij}^{\mathrm{var}} = \mathrm{Var}(\mathrm{contrib}_{ij,t}).$$

Across all variants, scalar scores remain unable to distinguish among qualitatively different causal mechanisms. In particular, mechanisms exhibiting threshold activation, saturation, or sign reversal continue to receive similar scalar values under these alternative summaries. This suggests that the limitation is dimensional rather than metric-specific: any scalar aggregation necessarily collapses state-dependent structure.

### D.3. Interpretation

Taken together, these robustness checks support two conclusions:

1. Scalar causal scores are stable with respect to randomness and moderate changes in the data-generating process.

2. Stability does not imply adequacy: despite their robustness, scalar scores consistently fail to encode the functional form, regime dependence, and sign structure of causal influence.

These findings reinforce the central claim of the paper: the limitations of scalar causal summaries arise from their representational compression, not from noise, instability, or poor estimation.

## E. Democracy Data Preprocessing

To evaluate function-valued causal influence in a realistic, theory-driven setting, we draw on data from the Varieties of Democracy (V-Dem) project (Coppedge et al., 2025; Pemstein et al., 2025), one of the most comprehensive datasets for measuring democratic institutions and governance. V-Dem provides expert-coded, theoretically grounded indicators capturing multiple dimensions of democracy across countries and time, and is widely used in both methodological and substantive research.

We use version 15 of the V-Dem country-year dataset, which spans the period 1789–2024. From this corpus, we select 16 lower-level democracy components that serve as foundational inputs to V-Dem's higher-order democracy indices. These components are conceptually well-defined, empirically validated, and commonly employed in the construction of aggregate democracy measures. Focusing on lower-level components allows us to study causal interactions among institutional dimensions directly, rather than relying on pre-aggregated indices. Desription of the components used in analysis is presented in Table 3.

*Table 3.* Democracy components from the V-Dem dataset used in the empirical analysis.

| # | Component name | V-Dem variable |
|---|---|---|
| 1 | Freedom of expression and alternative information | `v2x_freexp_altinf` |
| 2 | Freedom of association (thick) | `v2x_frassoc_thick` |
| 3 | Suffrage (share of population with voting rights) | `v2x_suffr` |
| 4 | Clean elections | `v2xel_frefair` |
| 5 | Elected officials | `v2x_elecoff` |
| 6 | Equality before the law and individual liberty | `v2xcl_rol` |
| 7 | Judicial constraints on the executive | `v2x_jucon` |
| 8 | Legislative constraints on the executive | `v2xlg_legcon` |
| 9 | Civil society participation | `v2x_cspart` |
| 10 | Direct popular vote | `v2xdd_dd` |
| 11 | Local government elections | `v2xel_locelec` |
| 12 | Regional government elections | `v2xel_regelec` |
| 13 | Deliberative component | `v2xdl_delib` |
| 14 | Equal access | `v2xeg_eqaccess` |
| 15 | Equal distribution of resources | `v2xeg_eqdr` |
| 16 | Equal protection | `v2xeg_eqprotec` |

### E.1. Panel construction and missingness

We retain only countries with complete and balanced observations on all selected democracy components over the full analysis window. Countries exhibiting substantial missingness or discontinuous coverage are excluded rather than imputed. This choice avoids imputation-driven artifacts and ensures that subsequent resampling procedures, such as block-based bootstrap inference, and out-of-sample

forecasting remain valid.

After applying these filters, the final dataset consists of 139 countries, each observed for 35 consecutive years (1990–2024), forming a strongly balanced panel. This structure is well suited for causal time-series modeling, as it preserves temporal dependence, cross-variable feedback, and comparability across units.

### E.2. Preprocessing and normalization

All democracy variables are standardized to zero mean and unit variance prior to model estimation. Standardization improves numerical stability during training and ensures that intervention-based analyses, such as ICE, operate on a common scale across variables. No additional smoothing or detrending is applied, allowing the models to learn temporal dynamics directly from the observed data.

### E.3. Suitability for causal analysis

The resulting dataset provides a challenging but informative testbed for nonlinear causal discovery. The panel exhibits strong persistence, cross-variable dependence, and heterogeneous dynamics across countries, reflecting the complexity of real-world institutional change. As such, it allows us to assess whether scalar causal summaries adequately capture learned causal structure, and whether function-valued causal influence provides additional insight in realistic settings.

### E.4. Correlation Structure of Democracy Components

All components are lower-level indices that serve as building blocks for higher-order democracy measures in V-Dem and are treated as endogenous variables in the causal time-series analysis. Table 4 presents pairwise Pearson correlations of the components, originally calculated on a standard $[0-1]$ scale.

*Table 4.* Pairwise Pearson correlations among democracy components (1990–2024).

| | Expr | Assoc | Suff | Clean | Elect | Liberty | Judic | Legisl | Civil | Direct | Local | Regional | Delib | Access | Distrib | Protect |
|---|---|---|---|---|---|---|---|---|---|---|---|---|---|---|---|---|
| Freedom of expression | 1.00 | | | | | | | | | | | | | | | |
| Freedom of association | 0.92 | 1.00 | | | | | | | | | | | | | | |
| Suffrage | 0.22 | 0.30 | 1.00 | | | | | | | | | | | | | |
| Clean elections | 0.77 | 0.74 | 0.21 | 1.00 | | | | | | | | | | | | |
| Elected officials | 0.51 | 0.58 | 0.29 | 0.42 | 1.00 | | | | | | | | | | | |
| Individual liberty | 0.86 | 0.83 | 0.17 | 0.82 | 0.45 | 1.00 | | | | | | | | | | |
| Judicial constraints | 0.80 | 0.77 | 0.10 | 0.80 | 0.31 | 0.83 | 1.00 | | | | | | | | | |
| Legislative constraints | 0.82 | 0.75 | 0.17 | 0.78 | 0.36 | 0.78 | 0.86 | 1.00 | | | | | | | | |
| Civil society participation | 0.87 | 0.82 | 0.21 | 0.70 | 0.40 | 0.79 | 0.77 | 0.78 | 1.00 | | | | | | | |
| Direct popular vote | 0.19 | 0.18 | 0.10 | 0.20 | 0.18 | 0.18 | 0.09 | 0.14 | 0.14 | 1.00 | | | | | | |
| Local government elections | 0.58 | 0.58 | 0.16 | 0.55 | 0.35 | 0.49 | 0.51 | 0.53 | 0.57 | 0.17 | 1.00 | | | | | |
| Regional government elections | 0.38 | 0.38 | 0.11 | 0.39 | 0.24 | 0.32 | 0.33 | 0.35 | 0.37 | 0.13 | 0.43 | 1.00 | | | | |
| Deliberative component | 0.85 | 0.78 | 0.17 | 0.76 | 0.41 | 0.81 | 0.80 | 0.83 | 0.84 | 0.12 | 0.49 | 0.33 | 1.00 | | | |
| Equal access | 0.60 | 0.53 | 0.12 | 0.61 | 0.29 | 0.75 | 0.60 | 0.58 | 0.57 | 0.20 | 0.30 | 0.19 | 0.61 | 1.00 | | |
| Equal distribution | 0.74 | 0.69 | 0.24 | 0.72 | 0.39 | 0.76 | 0.71 | 0.69 | 0.72 | 0.23 | 0.49 | 0.33 | 0.73 | 0.75 | 1.00 | |
| Equal protection | 0.34 | 0.27 | -0.05 | 0.59 | 0.12 | 0.58 | 0.50 | 0.45 | 0.31 | 0.17 | 0.19 | 0.10 | 0.45 | 0.68 | 0.58 | 1.00 |

The correlation matrix highlights strong persistence and cross-variable dependence among most democracy components, underscoring the challenge of interpreting scalar causal summaries in such settings.

### E.5. Implied NAVAR Causal Graph

Using the NAVAR training procedure described in Appendix A on the study's democracy dataset, we obtained the NAVAR implied causal graph shown in Figure 5.

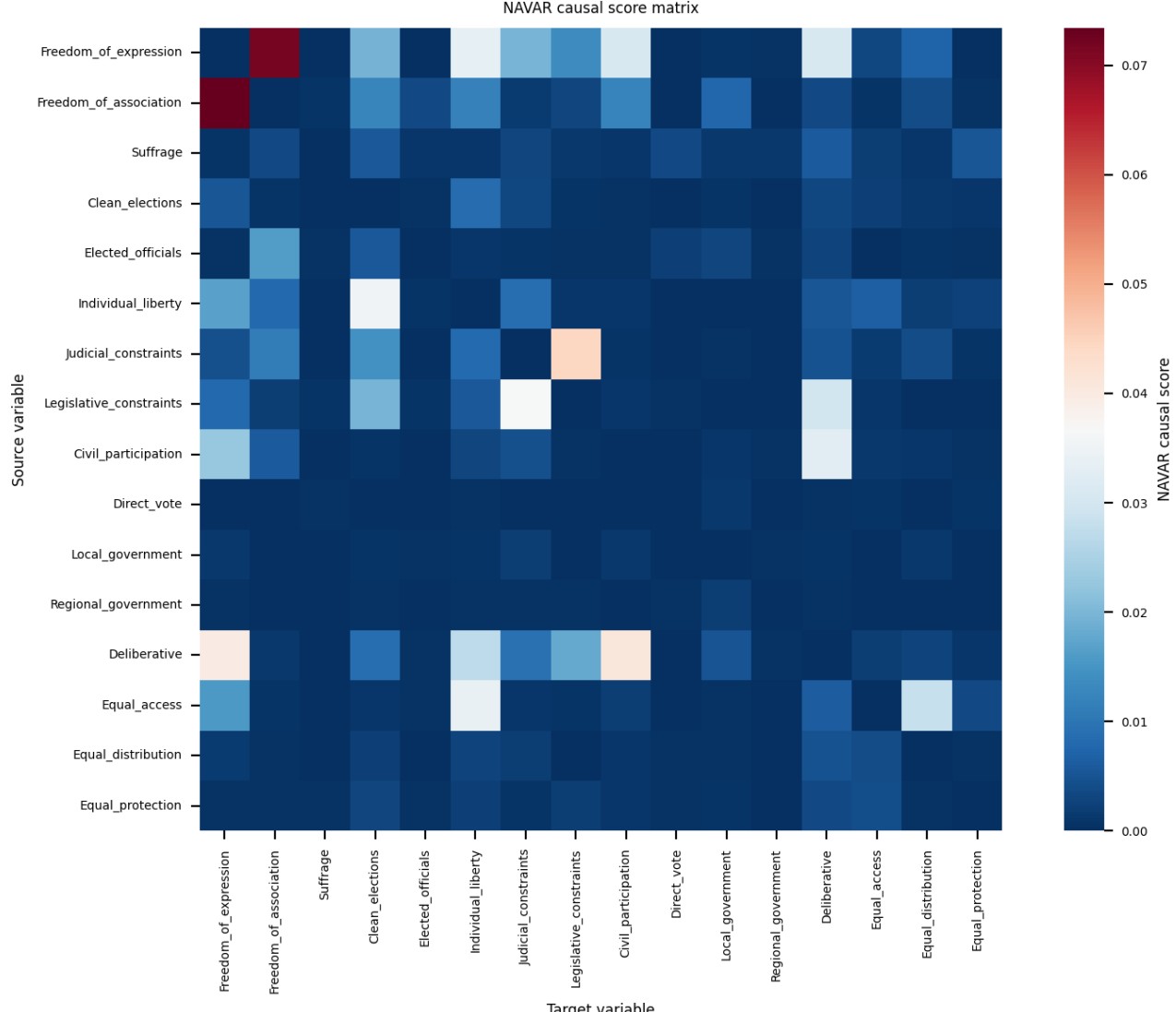

*Figure 5.* NAVAR Causal Score Matrix computed on democracy dataset. Each scalar in the matrix represents the variability (standard deviation) of the learned contribution from source $i$ to target $j$ across all time windows and answers the question "How strongly does $i$ help predict future values of $j$, across the entire time period?"

## F. ICE as a Conditional Expectation: Remark

In additive autoregressive models, intervening on a source variable's lagged history and averaging the resulting prediction differences across time windows yields a Monte Carlo approximation to the conditional expectation defining $g_{ij}(x)$. No assumptions are required beyond those in training.

## G. Bootstrap Uncertainty Quantification for ICE Curves

This section reports bootstrap confidence intervals for the lag-aggregated ICE curves presented in Figure 3 of the main text, addressing the absence of uncertainty quantification for the estimated

response functions.

### G.1. Method

We construct pointwise confidence bands using a non-parametric bootstrap over panel windows. For each of $B = 200$ resamples, we draw $W$ windows with replacement from the full set of valid lag windows (preserving regime assignments), recompute the lag-aggregated ICE curves for each regime tertile, and collect the resulting $(3 \times G)$ response matrices. Pointwise 95% confidence intervals are then obtained as the 2.5th and 97.5th percentiles of the bootstrap distribution at each grid point. This procedure resamples at the window level rather than the unit level, which is appropriate given the panel structure: windows from the same country are correlated, but resampling windows is a conservative approximation that does not require explicit block structure.

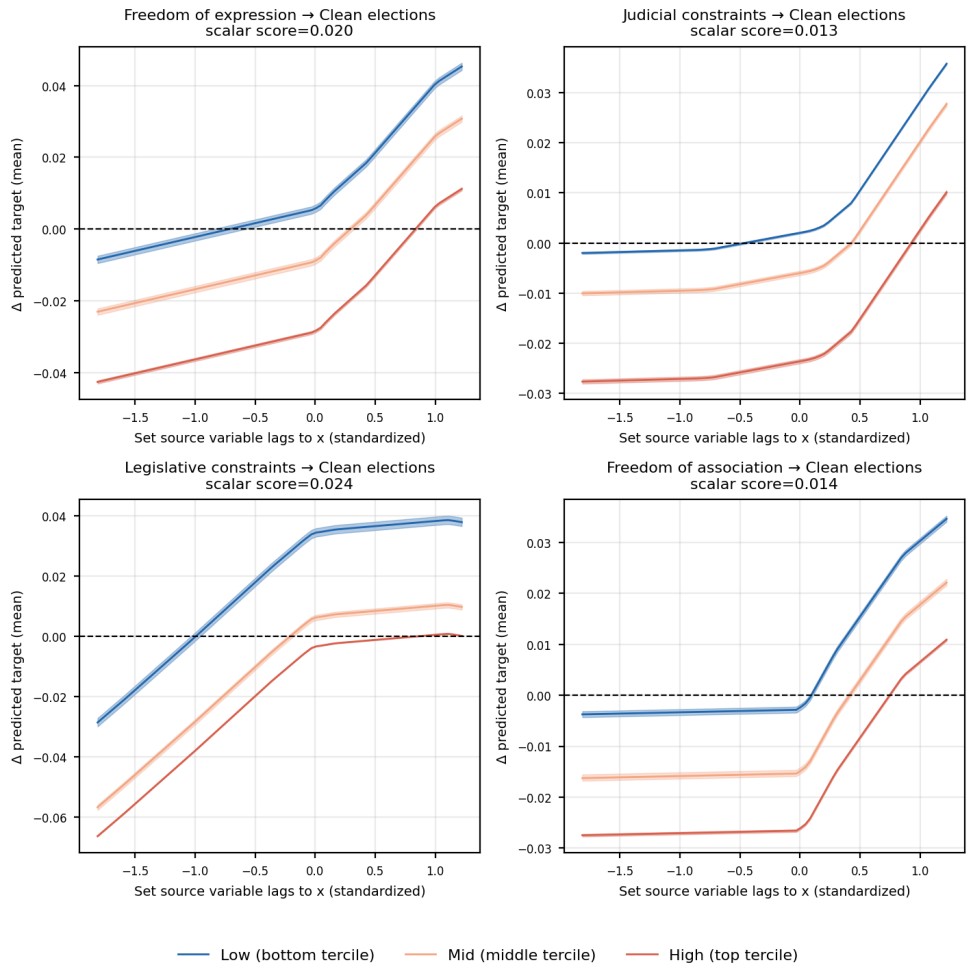

*Figure 6.* ICE response functions with 95% bootstrap confidence intervals (shaded bands; $B = 200$ resamples) for the four democracy edges shown in Figure 3. Narrow bands confirm that the estimated regime-conditional response functions are stable across sampling variability. The qualitative patterns - threshold activation, asymmetry, saturation, and regime reversal - are consistent across all bootstrap resamples, supporting the substantive interpretations in Section 5.2.

### G.2. Results

Figure 6 shows the four ICE response functions from Figure 3 with 95% bootstrap confidence bands (shaded regions). All four edges exhibit narrow, stable confidence bands across the full grid, confirming that the estimated response functions are robust to sampling variability and not artifacts of a specific draw of the panel data. The qualitative patterns - threshold activation, asymmetry, saturation, and regime reversal - are stable across bootstrap resamples, supporting the substantive interpretations offered in Section 5.2.

## H. Quantitative Recovery Quality in Synthetic Experiments

This section provides a quantitative complement to the visual recovery results shown in Figure 2, addressing the absence of scalar recovery metrics in the main text.

### H.1. Method

For each of the four synthetic causal mechanisms and each of 15 independent runs, we train NAVAR on the generated dataset, estimate the lag-aggregated ICE curve $\hat{g}_{XY}^{\mathrm{agg}}(x)$ on an 81-point grid spanning the 2nd to 98th percentile of the observed source variable, and compute the Pearson correlation between the estimated curve and the true mechanism $g(x)$ evaluated on the same grid. The Pearson correlation measures directional fidelity — whether the estimated curve correctly captures the shape of the true causal function — independently of scale differences introduced by model normalization.

### H.2. Results

Table 5 reports the mean, standard deviation, minimum, and maximum Pearson $r$ across 15 runs for each mechanism. All four mechanisms achieve Pearson $r \geq 0.968$, with standard deviations below 0.004, indicating high and stable recovery quality across seeds. Linear and saturating mechanisms achieve near-perfect recovery ($r = 0.996$), while threshold and sign-changing mechanisms — which involve discontinuities or sharp reversals — achieve slightly lower but still high fidelity ($r = 0.968$ and $r = 0.977$, respectively). These results confirm that lag-aggregated ICE reliably recovers the qualitative functional form of the true causal mechanism, including nonlinear features such as thresholds and sign changes, even from finite and noisy panel data. Figure 7 visualizes the recovery quality across mechanisms.

*Table 5.* ICE recovery quality by causal mechanism: Pearson correlation between true $g(x)$ and estimated $\hat{g}_{XY}^{\mathrm{agg}}(x)$ on an 81-point evaluation grid, averaged across 15 independent runs. Higher values indicate better recovery of the functional form.

| Mechanism | Mean $r$ | Std | Min | Max |
|---|---|---|---|---|
| Linear | 0.996 | 0.002 | 0.991 | 0.999 |
| Saturating | 0.996 | 0.002 | 0.991 | 0.999 |
| Sign-changing | 0.977 | 0.002 | 0.975 | 0.980 |
| Threshold | 0.968 | 0.003 | 0.961 | 0.973 |

## I. Prevalence of Nonlinear Functional Form in the Democracy Panel

This section addresses the scope of functional heterogeneity across all democracy edges, complementing the four selected edges analyzed in Section 5.2 with a broader quantitative survey.

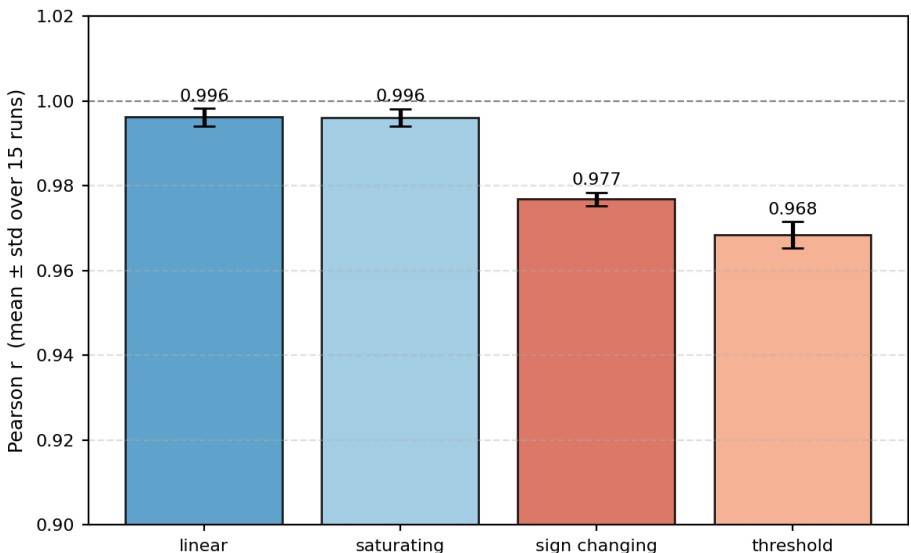

*Figure 7.* ICE recovery quality by causal mechanism, measured as Pearson correlation between the true $g(x)$ and the estimated $\hat{g}_{XY}^{\mathrm{agg}}(x)$ on an 81-point evaluation grid. Bars show mean $\pm$ std across 15 independent runs. All mechanisms achieve $r \geq 0.968$, confirming high recovery fidelity across functional forms.

## I.1. Method

We apply the lag-aggregated ICE framework to all directed edges $(i \rightarrow j)$ in the democracy panel with NAVAR causal score above a minimum threshold ($\hat{S}_{ij} \geq 0.005$), yielding $n = 50$ edges for analysis. For each edge, we estimate regime-conditional ICE curves across three tertile bins of the target variable's lagged value and classify the resulting response function according to four indicators:

- **Monotone:** the aggregate curve has fewer than 10% monotonicity violations across grid steps.

- **Threshold activation:** the low-regime response magnitude is less than 40% of the high-regime response magnitude, indicating activation above a minimum baseline.

- **Saturating:** either tail of the aggregate curve has a range less than 15% of the middle section's range, indicating diminishing marginal influence.

- **Regime reversal:** the aggregate curve crosses zero, indicating that the direction of causal influence reverses across levels of the source variable.

These indicators are not mutually exclusive: a curve can be simultaneously monotone and saturating, or exhibit both threshold activation and regime reversal.

## I.2. Results

Table 6 and Figure 8 summarize the prevalence of each indicator.

Every edge above the score threshold exhibits at least one nonlinear functional form indicator. The most prevalent indicator is regime reversal (98% of edges): the direction of causal influence between democratic institutional components reverses across low- and high-democracy contexts. This is a substantively important finding — it means that institutional relationships that appear stabilizing in

*Table 6.* Prevalence of nonlinear functional form indicators across 50 democracy edges with NAVAR score $\geq 0.005$. Categories are not mutually exclusive.

| Indicator | Count | % of edges |
|---|---|---|
| Monotone | 43 | 86% |
| Threshold activation | 5 | 10% |
| Saturating | 16 | 32% |
| Regime reversal | 49 | 98% |
| Any nonlinear indicator | 50 | 100% |

high-democracy countries may operate differently or even in the opposite direction in lower-democracy contexts. Saturating behavior is present in 32% of edges, consistent with the ceiling effects expected among high-democracy countries. Threshold activation, the sharpest form of nonlinearity, is present in 10% of edges. These patterns are entirely invisible to scalar causal scores, which collapse all functional heterogeneity into a single magnitude and cannot distinguish monotone from non-monotone, threshold-activated from globally active, or regime-reversing from unidirectional causal relationships.

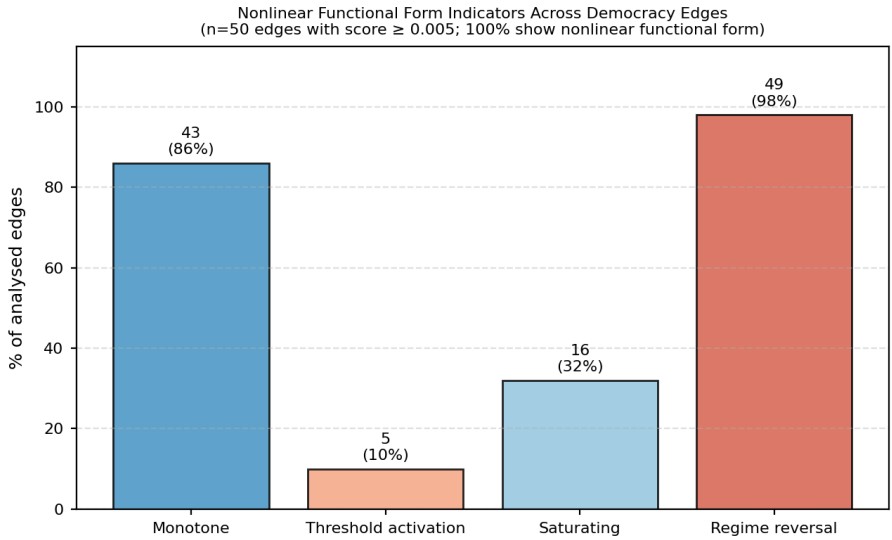

*Figure 8.* Prevalence of nonlinear functional form indicators across 50 democracy edges with NAVAR score $\geq 0.005$. All edges exhibit at least one nonlinear indicator. Direction of causal influence reversing across low- and high-democracy contexts (regime reversal) is the most prevalent pattern (98%), followed by saturation (32%) and threshold activation (10%). These patterns are invisible to scalar causal scores.

