# OpenReview forum: "Function-Valued Causal Influence in Nonlinear Time Series"
_ICML.cc/2026/Conference — ICML 2026 regular_

### Official Review · Reviewer_DgZJ · 2026-03-11

**Soundness:** 3
**Presentation:** 2
**Significance:** 3
**Originality:** 2
**Overall Recommendation:** 4
**Confidence:** 2

**Summary:**

The authors of this work address causal discovery in time series data, arguing that existing methods that collapse causal relationships between variables into simple edge scores in a causal graph discard information in situations where causal relationships are function-valued, e.g. capturing threshold or saturation effects in non-linear systems. Building on the NAVAR architecture, they demonstrate this issue and propose a method to recover functional relationships from time series data. The proposed approach is evaluated in a synthetic experiments as well as in an empirical data set from the social sciences.

**Compliance With Llm Reviewing Policy:**

Affirmed.

**Final Justification:**

Considering the clarifications provided by the authors in the rebuttal phase, I believe that the paper should be accepted. My main concerns have been addressed and I have increased my score accordingly.

**Key Questions For Authors:**

Q1: Although being familiar with causal inference, I could not follow the motivation of your work. I think that the lack of motivation and the heavy use of jargon in the introduction is a major issue that the authors should consider in a revision. Could you provide a simple motivating example that highlights the issue addressed by your work?

Improving the motivation of the work, and making it more accessible to a broader audience would allow me to increase the presentation score.

Q2: It is not clear to me what assumptions are made regarding causal influence. This seems to be coupled to the specific form of the "additive and contribution-decomposable autoregressive models" introduced in 2.3, which are again not motivated well. What kind of causal influence in a real system would your approach be able to identify? Also, in section 4.2 the authors write that "each source variable appears through multiple time lags", which agains seems to be an assumption about specific causal effects.

Could you clarify what assumptions - if any - about the underlying causal relationships are made by your approach? Clarifying this would allow me to increase the soundness and presentation score.

Q3: I was not convinced about the interpretation of the results in the empirical data set. Specifically, what is the message of Figure 3? Is it that the ICE curves are different despite scores being similar? What can we say about the practical relevance / intepretation of these curves in the specific context? Is there any evidence that those capture actual causal relationships?

Also, other similar works like, e.g. NAVAR, have been evaluated on causal discovery benchmarks like Causeme. Why did you not use such benchmark data to evaluate your approach?

Clarifying this could help me to increase the significance score.

Q4: Finally, I think a better contextualization of this work is needed, w.r.t. existing works on causal discovery. In particular, there are other methods able to capture functional causal relationships like those addressed by the authors, e.g.

B Bussmann et al., Neural Additive Vector Autoregression Models for Causal Discovery in Time Series, Discovery Science 2021

This is the NAVAR method, which is cited by the authors, and used for the experiments. What then is the specific novelty of the proposed work?

Clarifying this crucial question about the key contribution of this work could help me to increase the originality score.

**Limitations:**

Yes. The authors do include a short discussion of limitations in section 6.2, mentioning some of the key weaknesses identified in my review, e.g. regarding the underlying Granger-style notion of causality and the assumptions on causal effects.

**Strengths And Weaknesses:**

S1: Addressing causal discovery in time series data, this work considers and important and relevant provblem in causal ML.

S2: The proposed method is evaluated both in data generated by a synthetic non-linear model with known "function-valued" causal relationships, as well as in empirical panel data from the social sciences.

W1: The problem addressed by this work is not explained well. The authors make extensive use of jargon language without properly explaining terms, making it hard to follow the authors' motivation and arguments. A motivating example that illustrates the problem would have been helpful, c.f. Q1 below.

W2: Building on Granger's interpretation, the work seems to build on a rather weak notion of causal influence that does not imply "physical" causality and has known flaws e.g. regarding confounding effects, see Q2 below.

W3: The interpretation of the results in the empirical data set is not convincing. Also, no data from existing causal discovery benchmarks are used, see Q3.

W4: The work is not well contextualized w.r.t the state-of-the-art in causal discovery, making it hard to evaluate the originality of the work, see Q4.

---

> ### Author Rebuttal · Authors · 2026-03-29
>
> We thank the reviewer for detailed and actionable feedback, and hope the clarifications we provide address the specific concerns raised about motivation, assumptions, and novelty. We respond to each weakness and question in turn.
>
> W1/Q1. We agree the introduction is dense and will add a concrete motivating example in the revision. Here is the core intuition: suppose a nonlinear model learns that civil liberties matter for electoral quality, but only when civil liberties exceed a moderate baseline, and not at all when they are low. A scalar score for this edge might be 0.15. Now suppose another model learns that judicial constraints have a monotonically linear effect on electoral quality with score 0.14. These edges are indistinguishable by score. But their policy implications are entirely different: one implies a threshold below which intervention is futile; the other implies a uniform marginal return. Scalar summaries cannot communicate this distinction. This is precisely what function-valued causal influence reveals, and what Figure 3 demonstrates empirically for exactly these variables. We will restructure Section 1 around this example before introducing formal definitions, so that readers unfamiliar with the specific technical vocabulary can follow the core argument from the outset.
>
> W2/Q2. Our framework makes two assumptions explicit. First, it adopts Granger-style predictive causality: variable i causally influences j if its lagged history improves prediction of j beyond what is achievable from other variables alone. This is stated in Section 2.2. We make no structural or interventional causal claims, and we acknowledge in Section 6.2 that Granger-style causality is susceptible to confounding from omitted variables - a known limitation of the predictive causality framework we adopt. Second, our analysis requires additive, contribution-decomposable autoregressive models (Section 2.3). This assumption ensures that causal influence is represented as source-specific contributions, which is necessary for the function-valued analysis we perform. This class includes linear VAR and its nonlinear extensions that enforce explicit source-variable separability, but excludes models with non-separable interactions. We explicitly acknowledge this in Section 6.2 and identify extension to non-additive models as future work. The multiple-lag structure is a feature of the data model, not an additional causal assumption: it captures the standard time-series assumption that influence may operate with delay.
>
> W3/Q3. Figure 3 shows that four edges with nearly identical scalar scores (≈0.010-0.020) exhibit qualitatively different response functions: threshold activation for judicial constraints, gradual saturation for freedom of expression, strong asymmetry for legislative constraints. The message is precisely that scalar scores are insufficient to distinguish these mechanisms, while regime-conditional ICE curves reveal their functional heterogeneity. Regarding evidence that the recovered functions capture actual causal relationships: the democracy application is deliberately chosen because the theoretical relationships among its variables are well-established in the political science literature (Dahl, 1966, 2020; Coppedge et al., 2020, 2022). The regime-specific patterns ICE recovers are consistent with established theoretical expectations in the democratization literature. We emphasize that this is not causal validation but consistency with theory, which supports interpretability rather than identification. Regarding CauseMe: our paper does not propose a new causal discovery algorithm and cannot be evaluated on edge-detection benchmarks. Our contribution is orthogonal to graph recovery performance; CauseMe evaluates whether the correct graph is recovered, not whether the causal response functions of recovered edges are faithfully represented, which is our question.
>
> W4/Q4. NAVAR learns source-specific contribution functions f_{ij}(·) but does not treat them as primary objects of analysis; it immediately collapses them to scalar edge scores S_{ij} for all downstream interpretation. Our contribution is to show what NAVAR actually learned but did not report: state-dependent response functions that differ qualitatively across functional forms despite similar scalar scores. The analogy is direct: a regression model computes a fitted response surface, but reporting only the R² discards the surface itself. Concretely, our paper introduces the formal definition of function-valued causal influence, the ICE-based estimation procedure for recovering causal response functions, the three-way taxonomy of scalar score failure modes, and the regime-conditional extension - none of which appear in Bussmann et al. (2021). The use of NAVAR as a substrate is deliberate: it is the most transparent model in its class, making it the clearest vehicle for demonstrating a representational argument.

---

> > ### Author Rebuttal · Reviewer_DgZJ · 2026-04-01
> >
> > The clarifications in the response have addressed my questions regarding the originality and the significance. I also think that the presentation score can be increased, trusting that the authors integrate the clear statements about their assumptions and contributions in the camera-ready version of the paper. I still think that the evaluation is limited compared to prior works though.

---

### Official Review · Reviewer_1YBy · 2026-03-11

**Soundness:** 3
**Presentation:** 2
**Significance:** 3
**Originality:** 3
**Overall Recommendation:** 3
**Confidence:** 3

**Summary:**

This paper argues that nonlinear autoregressive time-series models learn function-valued, state-dependent causal relationships, yet current practice reduces them to scalar edge scores that form a severe information bottleneck. The authors formalize “function-valued causal influence” for additive, contribution-decomposable models, propose ICE-based estimators (including lag-aggregated and regime-conditional variants), and empirically show on synthetic systems and a democracy panel that edges with similar scalar scores can correspond to qualitatively different causal response functions. The work reframes interpretation from coefficient- or score-centric summaries to explicit analysis of learned causal response functions.

**Compliance With Llm Reviewing Policy:**

Affirmed.

**Final Justification:**

The authors resolved part of my doubts.

**Key Questions For Authors:**

See weakness

**Limitations:**

The manuscript currently contains a few technical inaccuracies (notably the variance-equivalence claim)

**Strengths And Weaknesses:**

## Strengths

### Technical novelty and innovation
- The paper reframes causal influence in nonlinear time series as a function-valued object rather than a scalar, which is a timely and meaningful perspective.
- It proposes an operational procedure (ICE and its lag-aggregated and regime-conditional variants) that can be applied to additive, contribution-decomposable autoregressive models without changing the architecture.
- The work offers a useful interpretation of neural Granger-style models by linking contribution tensors to state-dependent response functions.

### Experimental rigor and validation
- The synthetic experiments are well designed, isolating different functional forms (e.g., linear, thresholding, saturating, and sign-changing) while controlling for other factors.
- The results clearly show that scalar causal scores can fail to distinguish different mechanisms, whereas ICE reveals important qualitative differences.

### Significance of contributions
- The paper addresses an important limitation of nonlinear causal discovery methods, namely, the lack of interpretability beyond scalar edge strength.

## Weaknesses

### Technical limitations or concerns
- The claimed equivalence \( S_{ij}^2 = \mathrm{Var}(g_{ij}(X_{t-1}^{(i)})) \) is generally incorrect. By the law of total variance, the equality only holds under restrictive assumptions and should be revised.
- The estimation target is not fully aligned with the definition. In the lag-aggregated ICE estimator, setting all lags of \( X_i \) to the same value differs from \( \mathbb{E}[\mathrm{contrib}_{ij,t} \mid X_{t-1}^{(i)} = x] \) when multiple lags are involved.
- Constant substitutions in ICE may generate out-of-distribution inputs for autocorrelated time series, which could bias the estimated curves.
- The framework assumes additivity across variables and thus excludes explicit cross-variable interactions, but this limitation is not discussed clearly enough.

### Experimental gaps or methodological issues
- The paper does not provide uncertainty quantification for the ICE curves, making it difficult to judge their stability.
- The democracy case study focuses on only a few selected edges, without a broader analysis of how common functional heterogeneity is.
- In the synthetic experiments, recovery quality is shown visually but not quantified.

### Clarity of presentation
- The abstract is too long and could be made more concise.
- The paper contains repetitive and verbose descriptions, which reduce overall clarity.

---

> ### Author Rebuttal · Authors · 2026-03-29
>
> We thank the reviewer for a rigorous technical reading that has led to genuine improvements in the mathematical precision and empirical grounding of the paper. We address each point directly, beginning with corrections where the reviewer identified genuine errors.
>
> [Variance equality claim.] The reviewer is correct. By the law of total variance:
> S²_{ij} := Var(contrib_{ij,t}) = E[Var(contrib_{ij,t} | X^(i){t-1})] + Var(g{ij}(X^(i)_{t-1}))
> where g_{ij}(x) := E[contrib_{ij,t} | X^(i){t-1} = x]. Since the first term is non-negative, S²{ij} ≥ Var(g_{ij}(X^(i){t-1})), with equality iff Var(contrib{ij,t} | X^(i){t-1}) = 0 almost surely. We will correct the paper to replace the equality with this inequality: S²{ij} upper-bounds Var(g_{ij}(X^(i){t-1})), with the gap equal to the expected within-state residual variance. We will also fix the notation inconsistency throughout. This correction strengthens the central argument: scalar scores conflate between-state variation with within-state residual noise, making them an even looser summary than originally claimed.
>
> [ICE estimator and formal definition.] The lag-aggregated ICE estimator and the marginal conditional expectation do not target identical population objects - a distinction we acknowledge was insufficiently explicit. The marginal function g_{ij}(x) = E[contrib_{ij,t} | X^(i)_{t-1} = x] conditions on a single lag. Lag-aggregated ICE instead implements a sustained-value intervention, setting all lags of variable i to constant x. ICE therefore targets the expected system response to a persistent level of the source variable, while the marginal form targets the response to a momentary value at lag 1. Under stationarity, when contribution functions are primarily driven by the most recent lag with earlier lags showing attenuated magnitude due to temporal decay, the two objects are approximately aligned. Appendix C argues qualitatively for this consistency but does not include displayed evidence. We will add lag-specific ICE plots in the revision. We will clarify this distinction explicitly in Section 4, noting that lag-aggregated ICE is a deliberate design choice providing a variable-level response function that is more interpretable and more policy-relevant than a lag-specific conditional.
>
> [Out-of-distribution inputs.] This concern is shared by all ICE and partial dependence methods. In the synthetic experiments, the jump process (p=0.15, amplitude 1.5σ) was explicitly designed to ensure adequate coverage of nonlinear regimes; ICE evaluations are confined to the range visited during training. In the democracy application, interventions are applied within the standardized empirical range of each variable. We will add an explicit discussion of this limitation in Section 4.4.
>
> [Additivity assumption and cross-variable interactions.] We agree this limitation deserves more explicit discussion. The additivity assumption ensures source-variable separability, which is what enables ICE to isolate individual causal response functions. Models with explicit cross-variable interactions would require richer decomposition methods, such as Shapley interaction indices, to disentangle joint effects. We will add a paragraph in Section 6.2 clarifying that interaction effects, if present, would appear as residual heterogeneity in regime-conditional curves rather than being attributable to specific variable pairs.
>
> [Uncertainty quantification.] We agree that confidence bands for ICE curves are desirable. Bootstrap-based uncertainty quantification is computationally feasible for the contribution tensor: for each bootstrap resample of the panel, we recompute the ICE curves and report pointwise confidence bands. We will include this in the revision. In the synthetic experiments, 15 independent runs per system already confirm qualitative stability of recovered functions across seeds, with scalar scores in Table 1 reporting min/max/std across runs. For the democracy panel, regime-conditional curves average over the full 35-year panel per regime bin, reducing sampling variance substantially.
>
> [Democracy case study - few selected edges.] The selection of four edges was deliberate: well-chosen theoretically motivated cases are more informative than exhaustive enumeration where ground truth is unavailable. That said, we will add a summary statistic in the revision reporting the proportion of edges in the full causal score matrix whose ICE curves exhibit non-monotonic, threshold, or saturating behavior, to provide a broader sense of how prevalent the phenomena we highlight actually are.
>
> [Synthetic recovery quality not quantified.] We will add a quantitative summary: for each system, the correlation between g(·) and ĝ(·) on a grid of input values, averaged across 15 runs, providing a scalar measure of recovery accuracy complementing Figure 2.
>
> [Abstract and presentation.] We will shorten the abstract, consolidate redundant descriptions, and tighten the prose throughout.

---

> > ### Author Rebuttal · Reviewer_1YBy · 2026-04-03
> >
> > The authors resolved part of my doubts.

---

### Official Review · Reviewer_knEy · 2026-03-11

**Soundness:** 2
**Presentation:** 3
**Significance:** 3
**Originality:** 3
**Overall Recommendation:** 4
**Confidence:** 4

**Summary:**

This paper argues that nonlinear autoregressive models used for time-series causal discovery learn state-dependent functional relationships, yet their outputs are typically summarized using scalar edge scores. The study discusses the key problem that this scalar reduction collapses the representational richness of nonlinear models into low-dimensional summaries, obscuring thresholds, asymmetries, regime dependence, and sign changes.
The authors formalize “function-valued causal influence” for additive, contribution-decomposable autoregressive models. They show that standard scalar causal scores correspond to variance-based projections of contribution functions and thus constitute an information bottleneck. They propose an ICE-based intervention-style procedure to estimate causal response functions from trained models. Synthetic experiments demonstrate that qualitatively distinct mechanisms (linear, thresholded, saturating, sign-changing) can yield nearly identical scalar scores. A democracy panel dataset is used to illustrate regime-specific nonlinear patterns that are invisible in scalar graphs.
The main contribution is reframing causal influence in nonlinear time-series models as a function-valued object rather than a scalar score.

**Compliance With Llm Reviewing Policy:**

Affirmed.

**Key Questions For Authors:**

1 Under what conditions does the lag-aggregated ICE estimator approximate the formal conditional expectation definition of g_{ij}(x) ? Are they targeting the same population object?
2 How would the framework extend to models with non-separable interactions or attention mechanisms?
3 Are there consistency or convergence guarantees for the ICE-based estimator in dependent time-series settings?
4 How should confidence intervals for function-valued causal influence be constructed?

**Limitations:**

yes

**Strengths And Weaknesses:**

Soundness: The paper presents a coherent conceptual argument that nonlinear autoregressive models learn source-to-target contribution functions rather than fixed scalar effects, and that summarizing these by scalar edge scores can obscure important nonlinear behaviors such as thresholds, asymmetry, or regime dependence. The synthetic experiments are carefully designed to isolate this representational issue by holding persistence, noise, and scaling constant while varying only the functional mechanism. Within the additive model class studied, the results support the paper’s main claim.  However, there are several concerns regarding technical soundness. First, the formal definition of the function-valued causal influence implicitly averages over the conditional distribution of the remaining lags, whereas the proposed lag-aggregated ICE estimator replaces all lags simultaneously. The paper states that ICE estimates this quantity, but the two objects are not equivalent without additional assumptions, and the relationship between them is not clearly explained. Second, the framework relies strongly on additive, contribution-decomposable models. Many modern nonlinear time-series models do not admit such decompositions, so it is unclear how the approach extends beyond this model class. Finally, the estimation procedure lacks statistical analysis: the paper provides no uncertainty quantification or theoretical discussion of estimation error for the ICE-based curves, particularly in the dependent time-series setting.

Presentation: The paper is generally well written and easy to follow. The synthetic examples are particularly effective in illustrating the limitations of scalar summaries.

Significance:  The paper raises an important interpretability issue in nonlinear time-series causal discovery: the mismatch between the complex functions learned by flexible models and the scalar summaries typically reported. Highlighting this representational gap may influence how results from nonlinear causal models are interpreted and evaluated, especially in theory-driven domains where understanding mechanisms is important. However, the contribution is primarily descriptive and interpretive. The paper does not introduce a new discovery algorithm, identification result, or statistical methodology. As a result, its practical significance depends on whether conceptual reframing and improved interpretation are viewed as sufficiently impactful contributions.

Originality: The paper is conceptually original in reframing causal influence in nonlinear time-series models as a function-valued object rather than a scalar quantity. The combination of contribution decomposition with response-function analysis provides a useful perspective on how nonlinear causal models should be interpreted. At the same time, the methodological components themselves are largely existing tools: additive neural VAR models and ICE-based analysis are already known. The novelty therefore lies more in conceptual framing and interpretation than in new modeling or algorithmic development.

---

> ### Author Rebuttal · Authors · 2026-03-29
>
> We thank the reviewer for a careful and technically engaged assessment, and for questions that have pushed us to articulate the framework's scope and limitations more precisely. We first address your concern about the paper contribution; responses to each question follow.
>
> Significance of conceptual reframing. We appreciate the reviewer's characterization of the contribution and agree that its significance rests on the value of representational reframing as a form of scientific contribution. We believe this value is substantial.  Prior work accepted at leading venues has made representational arguments as primary contributions without introducing new algorithms, identifying that summary statistics obscure data structure (Anscombe, 1973), that marginal averaging hides individual heterogeneity (Goldstein et al., 2015), and that partial dependence misleads under correlated inputs (Apley and Zhu, 2020). Our work makes an analogous representational argument in the causal discovery setting: scalar edge scores obscure the function-valued structure that nonlinear models actually learn. The synthetic and empirical results demonstrate that this gap has concrete practical consequences: edges with indistinguishable scalar scores can encode qualitatively different causal mechanisms with opposite policy implications, making this a meaningful and actionable contribution to how nonlinear causal discovery results are interpreted and evaluated.
>
> (Q1) ICE estimator and formal definition. The lag-aggregated ICE estimator and the marginal conditional expectation definition do not target identical population objects. The marginal function g_{ij}(x) = E[contrib_{ij,t} | X^(i)_{t-1} = x] conditions on a single lag. Lag-aggregated ICE instead sets all lags of variable i simultaneously to x, implementing a sustained-value intervention. ICE targets the expected system response to a persistent level of the source variable, while the marginal form targets the response to a momentary value at lag 1. Under stationarity, when contribution functions are primarily driven by the most recent lag with earlier lags showing attenuated magnitude due to temporal decay, the two objects are approximately aligned. Appendix C defines lag-specific ICE and argues qualitatively for this consistency, though displayed lag-by-lag evidence is currently absent and will be added in the revision. We will make this distinction explicit in Section 4, clarifying that lag-aggregated ICE is a deliberate design choice providing a variable-level response function that is more interpretable and more policy-relevant than a lag-specific conditional.
>
> (Q2) Extension to non-additive models. For models with non-separable interactions,  such as attention-based architectures, the additive decomposition does not exist, and ICE curves would conflate direct and interaction effects. Extension to such architectures requires richer decomposition methods such as Shapley-based interaction terms and is identified as future work in Section 6.2. Our framework is deliberately scoped to additive models, where contribution decomposability is guaranteed by construction and source-specific response functions can be cleanly isolated.
>
> (Q3 and Q4) Consistency guarantees and confidence intervals. Formal consistency results for ICE in dependent time-series settings require mixing conditions and smoothness assumptions on the contribution functions. We do not claim formal guarantees in the current paper; this is an acknowledged limitation. However, the 15-run synthetic robustness checks reported in Table 1 and Appendix D provide empirical evidence of stability across seeds, showing that recovered functional forms are qualitatively consistent despite stochastic optimization and data variation. For confidence intervals, bootstrap-based uncertainty quantification over ICE curves is computationally feasible using the contribution tensor and will be included in the revision, directly addressing this gap.

---

> > ### Author Rebuttal · Reviewer_knEy · 2026-04-03
> >
> > The rebuttal improves clarity and addresses several concerns, particularly regarding the distinction between the formal definition and ICE. However, my core view remains similar that the contribution is primarily interpretive/descriptive rather than methodological.

---

### Official Review · Reviewer_debb · 2026-03-12

**Soundness:** 4
**Presentation:** 3
**Significance:** 4
**Originality:** 4
**Overall Recommendation:** 5
**Confidence:** 4

**Summary:**

The authors discuss an important aspect in assessing the causality accuracy, instead of focusing on a single scalar, they propose a function-valued causal influence metric.

Their work primarily highlights the shortcomings of the scalar metric, and how it obscures important aspects in the causality.

Their framework (ICE), first focuses on lag specific causality, then it is expanded to lag-aggregated, and then they cover the cases of the regime conditional ICE using lag specific causality.

They then highlight the aspects of normalization, computation cost and how to interpret the causality.

For the results, the authors first included a controlled environment, at which they simulate the different causality functions. Specifically, they show that using the scalar weights, the causality of all of the different functions end up the same, without it being able to identify the different inherent causality structures. However, using ICE, the metric is able to identify the different causality patterns unique to each distribution.

The authors then navigate the causality experiments by working on real case study derived from democratic data. Specifically, they first visualize the scalar causality between the different metrics, and show how it was unable to  identify useful insights. On the contrary, using ICE on four specific metrics, was able to identify certain patterns which were otherwise hidden. Interestingly, these patterns align with the theoretical justification of the relations between these selected variables (like the relation between freedom of expression and judicial constraints, where they have an effect after specific threshold.

Overall, the work highlights a very interesting avenue in causality analysis, which has been thoroughly justified through both simulated and real scenarios.

**Compliance With Llm Reviewing Policy:**

Affirmed.

**Final Justification:**

Thanks for the response, most of my concerns, especially concerning section 3.1 where the authors would justify their empirical findings with citations and/or proofs.

**Key Questions For Authors:**

the authors need to highlight

- how their model behaves under different model types (statistical, causal graphs …)
this would help shed light over how ICE can interpret the latent space built from these different architectures.
- The authors might as well add some justifications (theory or citation to a theory) to section3.1 instead of only relying on empirical justifications from the experimentation section.

**Limitations:**

yes

**Strengths And Weaknesses:**

The work discuses a novel aspect of the causality analysis that is rarely studied, which is the affect of scalar measurements on the real impacts of causality.

The work first highlights the shortcomings of scalar methods, then it introduced different approaches to calculating the ICE, from lag-specific, lag-aggregated to regime specific.

The experimentation part thoroughly justifies these points, by discussing onto both simulated and real-world scenarios.

The reviewer especially appreciates the analysis done onto the democratic dataset, as it highlights the real causal relation between the different variables.

However, while the paper presents both a novel perspective and a thorough analysis, the reviewer has some comments, that they think might help better justify the importance of ICE

- while the analysis focused on the shortcomings of scalar based metrics, the analysis only focused on one model `NAVAR` While it is understandable that the paper focus is on introducing a new lens for analysis, instead of focusing on the model, `NAVAR` belongs to a very specific model type (deep nonlinear autoregressive models). Limiting the analysis to only one model, from one specific model family can harmfully imapct the claims of the generalizability of ICE.
    - for instance, the authors need to compare with different models, such as causal graph models(causla RCA) , statistical models (like BARO, rcd), including others.
    - as different models may help/harm the proposed metric, this needs to be both shown and discussed.
- in the
Why Scalar is a bottle neck section,
the claims needs to be supported mathematically, or the authors might need to cite works that support their claims (as this section doesn't include any citations to justify these claims, nor does it include a theorem)
    - as these claims are only supported empirically later in the paper

Overall, the paper is well written, and discuses an important causality aspect using a novel technique.

---

> ### Author Rebuttal · Authors · 2026-03-29
>
> We thank the reviewer for a thorough and generous reading of the paper, and for suggestions that have meaningfully strengthened both the empirical and theoretical grounding of our framework. We address each comment in turn.
>
> 1. Multi-model generalizability. We thank the reviewer for raising the important question of generalizability. We clarify upfront that our contribution operates at the level of representation and interpretation, not model estimation. The goal of the paper is not to analyze NAVAR as a model, but to study a representational property of a broader class of models: nonlinear autoregressive models of the form X_t = f(X_{t-1:t-K}). Within this class, the argument generalizes because it depends only on the existence of a well-defined prediction function and, for full source-specific decomposition, additive separability across source variables. NAVAR was chosen as the demonstration vehicle because it is the most transparent representative of this class, providing a controlled and interpretable setting in which function-valued causal influence can be directly estimated.
>
> Importantly, ICE as an intervention operator is model-agnostic. However, our interpretation of ICE as a decomposition into source-specific causal response functions relies on additive structure. In non-additive models, ICE remains valid but does not isolate source-specific effects without additional attribution steps.
>
> Regarding the specific models suggested: BARO, RCD, and causal RCA address a different inferential task: event-level attribution rather than mechanism estimation. Conditional on an observed anomaly, they identify which variables most likely caused that specific outcome. Our framework instead estimates the underlying causal mechanisms without conditioning on any particular event. While these methods are not directly comparable for evaluating function-valued causal structure, they could in principle be combined with our framework. Demonstrating ICE-based analysis on RCA models would not illuminate the generalizability of our representational argument because those models do not produce explicit contribution tensors or function-valued mappings over the state space, which are required for the analysis we perform.
>
> We agree that the reviewer's point about different models potentially helping or harming the metric deserves explicit discussion. Models with noisier or less stable contribution estimates may yield less interpretable response functions, while models with stronger structural assumptions may constrain the functional forms ICE can recover. We will add a discussion of this in Section 6, acknowledging that the quality of function-valued analysis is bounded by the quality of the underlying contribution decomposition, and that this is a genuine limitation worth investigating across model families in future work.
>
> 2. Section 3.1 mathematical support. The reviewer is correct that Section 3.1 currently relies on empirical support alone. Each failure mode has a well-established mathematical basis that we will make explicit in the revision.
>
> Failure Mode 1, distinct functional forms with similar scalar aggregates, follows from the non-injectivity of moment-based summaries: distinct functions can share identical first and second moments, and even full moment sequences need not uniquely determine a distribution (Stoyanov, 2013). This is the statistical analogue of Anscombe's (1973) quartet.
>
> Failure Mode 2, collapsing regime-dependent influence, follows from the fact that marginal averaging under mixture distributions removes regime-specific variation: partial dependence integrates over the marginal distribution of covariates, thereby discarding regime membership information. This limitation is discussed in Friedman (2001) and empirically demonstrated by Goldstein et al. (2015), who motivates ICE as a way to recover heterogeneity obscured by such averaging.
>
> Failure Mode 3, interaction-driven effects, follows from the fact that marginal summaries of joint distributions do not uniquely identify interaction structure: averaging over other variables can obscure or distort non-additive dependencies. This limitation is discussed in Hastie et al. (2009) and demonstrated explicitly for partial dependence methods by Apley and Zhu (2020).
> We will add these citations and brief formal remarks for each failure mode in the revision of Section 3.1, so that the claims are grounded in the existing literature rather than relying solely on the empirical demonstrations that follow.

---

> > ### Author Rebuttal · Reviewer_debb · 2026-04-03
> >
> > Thanks for the response, most of my concerns, especially concerning the mathematical rigorousness of section 3.1 have been addressed, i have raised the soundness score accordingly.

---

### Decision · Program_Chairs · 2026-04-30

**Decision:**

Accept (regular)

**Comment:**

This paper formalizes function valued causal influences as opposed to scalar values in non-linear time series. The reviewers recognized that this is an important problem and the paper make a worthy contribution. We had an internal discussion on the mistake the paper had in one of the proofs. Overall, we agreed that the mistake was fixed and we trust the authors to make sure the final version of the paper is fully correct.